# Comparative transcriptomics of social insect queen pheromones

Luke Holman[1], Heikki Helanterä [2], Kalevi Trontti[3] & Alexander S. Mikheyev [4,5]

Queen pheromones are chemical signals that mediate reproductive division of labor in eusocial animals. Remarkably, queen pheromones are composed of identical or chemically similar compounds in some ants, wasps and bees, even though these taxa diverged >150MYA and evolved queens and workers independently. Here, we measure the transcriptomic consequences of experimental exposure to queen pheromones in workers from two ant and two bee species (genera: *Lasius*, *Apis*, *Bombus*), and test whether they are similar across species. Queen pheromone exposure affected transcription and splicing at many loci. Many genes responded consistently in multiple species, and the set of pheromone-sensitive genes was enriched for functions relating to lipid biosynthesis and transport, olfaction, production of cuticle, oogenesis, and histone (de)acetylation. Pheromone-sensitive genes tend to be evolutionarily ancient, positively selected, peripheral in the gene coexpression network, hypomethylated, and caste-specific in their expression. Our results reveal how queen pheromones achieve their effects, and suggest that ants and bees use similar genetic modules to achieve reproductive division of labor.

[1] School of BioSciences, University of Melbourne, VIC 3010 Melbourne, Australia. [2] Ecology and Genetics Research Unit, University of Oulu, FI-90014 Oulu, Finland. [3] Department of Biosciences, University of Helsinki, P.O. Box 65, Helsinki 00014, Finland. [4] Research School of Biology, Australian National University, Canberra, ACT 0200, Australia. [5] Ecology and evolution unit, Okinawa Institute of Science and Technology, Onna-son, Kunigami-gun, Okinawa 904–0412, Japan. Correspondence and requests for materials should be addressed to L.H. (email: luke.holman@unimelb.edu.au) or to A.S.M. (email: alexander.mikheyev@anu.edu.au)

Queen pheromones are chemical signals that characterise queens and other reproductive individuals in the social insects[1,2]. These signals can affect the behaviour of other colony members, e.g. by attracting workers[3], eliciting submissive behaviour[4], modulating aggression[5], or inhibiting production of new queens[6,7]. Queen pheromones also have long-lasting effects on individuals that encounter them, including reducing female fecundity[1], influencing the rate at which workers progress to different tasks with age[8], and altering workers' capacity to learn[9]. Queen pheromones are regarded as an honest signal of fecundity and condition, to which workers adaptively respond by continuing to express a worker-like phenotype, as opposed to a 'manipulative' adaptation that reduces worker fitness[10–12].

Current evidence suggests that most or all eusocial insects possess queen pheromones[1,2,12]. Although the eusocial bees, ants and wasps evolved eusociality (and thus, queen-worker communication) independently[13], most Hymenopteran queen pheromones are thought to be composed of chemically similar compounds[1,2]. Certain species of ants, wasps and bumblebees have been experimentally shown to use cuticular hydrocarbons (CHCs; a non-volatile blend of hydrocarbons adhering to the body surface) as queen pheromones, particularly certain long-chained alkanes, methylalkanes, and alkenes[1,2]. By contrast, honeybee queens (genus *Apis*) possess a pheromone composed of a blend of fatty-acid derived molecules (e.g. keto acids), which is secreted from the mandibular gland[3]. The similarity of non-*Apis* species' queen pheromones implies that these pheromones evolved from chemical cues or signals (e.g. sex pheromones) that were already present in the non-social most recent common ancestor of the social Hymenoptera[1,12].

Presumably, the profound changes in worker behaviour and physiology caused by queen pheromones stem from pheromone-mediated effects on the transcriptome. For example, queen pheromone exposure might stimulate or repress the expression of transcription or splicing factors, or affect their ability to bind to promoter regions and splice sites (e.g. by modulating epigenetic processes[14]). To our knowledge, only two previous studies—both using microarrays in the honey bee *Apis mellifera*—have experimentally measured the effects of queen pheromone on the whole transcriptome[15,16]. The response of other species to queen pheromones is unstudied, and so it is unclear whether different species use similar or distinct genetic pathways in queen-worker communication. The *Apis* research also bears repeating because only 18 of *c*. 1000 differentially expressed genes from the first study replicated in the second (three-fold fewer genes replicated than expected by chance alone[16]).

Here, we perform RNA sequencing on adult worker whole bodies to identify genes that are differentially expressed or alternatively spliced in response to experimental exposure to synthetic queen pheromones, in two bee and two ant species. Our first aim is to determine the extent to which pheromone-sensitive genes, pathways, and transcriptional modules are similar or distinct in ants and bees. The chemical similarity of some species' queen pheromones, coupled with the fact that queen pheromones influence similar phenotypic traits across the Hymenoptera[1], suggests that queen pheromones might affect many of the same genes across species. Conversely, we expect that some responses to pheromone will be unique to bees or to ants, given that these two taxa diverged over 150MYA, and independently evolved their eusocial societies (and thus, queen-worker communication). Second, we test whether queen pheromones influence alternative splicing. Alternative splicing is thought to mediate many insect polyphenisms, including the queen-worker polyphenism[17–20], but to our knowledge it is unstudied in relation to queen pheromones. Third, we aim to identify genes and pathways that respond to queen pheromone and thereby reveal how these key social signals produce their manifold phenotypic effects. Fourth, we test whether pheromone-deprived workers develop a more queen-like transcriptome[21] to match their queen-like phenotype (e.g. laying eggs and living longer[1,22]), thereby indirectly assisting the search for loci underlying caste dimorphism. Fifth, we test whether the genes affected by queen pheromones tend to be older than eusociality itself, which is interesting in light of the theory that chemical signalling of fecundity provided an important stepping stone to the eusociality[23].

## Results

**Effects of queen pheromone on gene expression.** Many genes showed statistically significant differential expression between the pheromone-treated and control groups in *A. mellifera* (322 genes), *L. flavus* (290), and *L. niger* (135), and a single gene was significant in *B. terrestris* (Fig. 1a; Supplementary Tables 2–5). The sets of significantly differentially expressed genes overlapped significantly more than expected for the two *Lasius* species (Fig. 1a; hypergeometric test: $p < 0.0001$). A smaller number of genes were significant in *Apis* and one *Lasius* species (Supplementary Table 6), though the number of overlaps was not significantly higher than expected under the null ($p = 0.19$ for *L. flavus* and $p = 0.27$ for *L. niger*). One gene was perturbed in 3/4 species: *myosin light chain alkali-like* (Supplementary Table 6).

Venn diagrams like those in Fig. 1 can give a misleadingly low impression of the numbers of pheromone-sensitive genes, since all studies have finite power to detect differential expression for any particular gene. Moreover, having finite power causes one to underestimate the number of genes that overlap between species, because detecting overlaps requires one to avoid multiple false negatives. For example, if our average power to detect a pheromone-sensitive gene in one species were 40%, we would only detect about $0.4^4 = 2.6\%$ of genes that were pheromone sensitive in all four species.

For this reason, we employed additional, better-powered analyses to test for conserved effects across species (see Methods). Pheromone sensitivity was significantly positively correlated across pairs of orthologous genes, for all possible species pairs (Fig. 1c,d, and Supplementary Fig. 3; all $p < 10^{-7}$). Thus, genes that were pheromone-sensitive in bees tended to also be pheromone-sensitive in ants. The cross-species correlations might be stronger than suggested by our figures, because the sensitivity of each gene is measured with error, which would obscure any underlying correlation.

When we ranked orthologous genes in order of their sensitivity to queen pheromone, there was some overlap between species in the top-*n*-ranked genes (Supplementary Table 8). For various *n*, six genes appeared in the top *n* most pheromone-sensitive genes for all four species: *serotonin receptor*, *protein takeout-like*, *titin-like*, *glucose dehydrogenase*, *histone-lysine N-methyltransferase SETMAR-like*, and *uncharacterized protein LOC102656088*. The amount of overlap in the top *n* genes was statistically significantly higher than expected for three species pairs, and marginally non-significant ($p < 0.08$) for an additional two species pairs, such that we found some evidence of non-random overlap for 5/6 species pairs (Supplementary Table 9).

The gene showing the single largest change in expression in *A. mellifera* was *Major Royal Jelly Protein 3*, which had 89-fold higher expression in workers deprived of queen pheromone (Supplementary Table 2). Indeed, in *Apis*, five out of the top 12 most differentially expressed genes were *Major Royal Jelly Protein 1, 2, 3,* and *4*, plus *major royal jelly protein 3-like*. As well as being biologically interesting, these results provide a validity check on our results, since *Apis* workers excrete royal jelly when rearing new queens, which they do when the queen (and her pheromone)

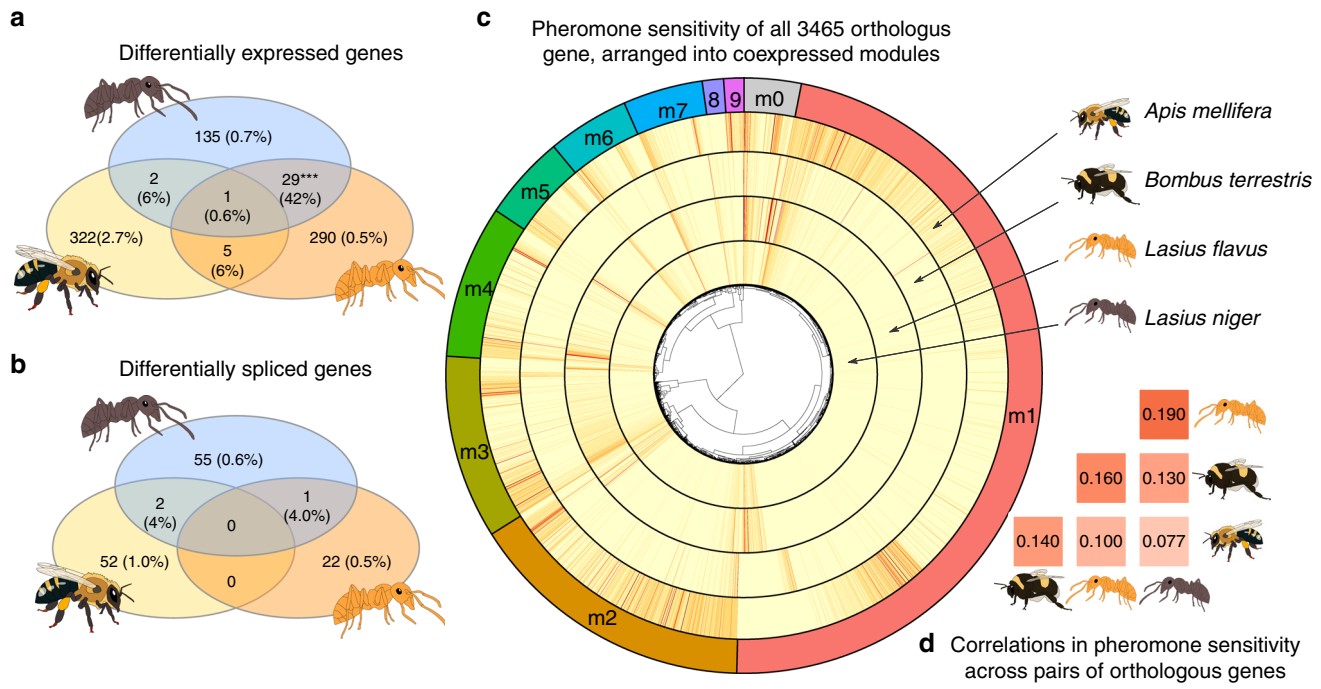

**Fig. 1** Summary of the effects of queen pheromone on gene expression and splicing, showing the extent of overlap between species. **a**, **b** Venn diagrams showing the number of significantly differentially expressed or differentially spliced genes per species ($\alpha = 0.05$ after false discovery rate correction). Parentheses in the outermost areas show this number as a percentage of all transcripts measured in the focal species, while parentheses in the inner areas show the number of overlapping genes as a percentage of the maximum number that *could* have overlapped (this number depends on the number of detectable orthologs and the number of significant genes). Asterisks denote the one overlap that was significantly higher than expected by chance (hypergeometric test, $p < 0.0001$). *B. terrestris* is omitted because there was only one significantly differentially expressed gene, and no differentially spliced genes. **c** Graphical overview of the amount of similarity in pheromone sensitivity in the set of 3465 orthologous genes. The four inner rings show the pheromone sensitivity of each gene (redder colours indicate increased sensitivity), and the genes have been clustered according to coexpression pattern, as shown in the central dendrogram. The coloured outer ring shows the assignment of genes to modules, and the grey area marked m0 refers to genes that were not assigned to a module. **d** Orthologous genes tended to show a similar level of sensitivity to queen pheromones for each pair of species. The numbers give Spearman's $\rho$ (Benjamini-Hochberg corrected $p < 10^{-7}$ in all cases)

are absent. Also, in *L. niger*, the second-most pheromone-sensitive gene was *Major Royal Jelly Protein 1*, which had 33-fold higher expression in controls (Supplementary Table 5), suggesting that queen pheromones affect this gene family in ants as well as bees.

**Gene set enrichment analysis of highly pheromone-sensitive genes.** Pheromone-sensitive genes (i.e. those showing a large difference in expression between treatments) were significantly enriched for many of the same GO (gene ontology) and KEGG (Kyoto Encyclopedia of Genes and Genomes) annotations across the four species (Fig. 2 and Supplementary Figs. 6–8, and Supplementary Table 18–S21). For example, the GO: Biological process term *defense response to bacterium* was significantly enriched in 3/4 species, and trended in the same direction for the fourth species (this result was driven by the pheromone sensitivity of genes like *defensin-1*, *hymenoptaecin*, and *apidaecin-1*). We also found 3/4 significant results for the GO: Molecular function terms *structural constituent of cuticle* (driven by a large number of cuticular proteins), *odorant binding*, and *olfactory receptor activity* (driven by many receptors and odorant binding proteins). Genes associated with the *extracellular region* (driven by the major royal jelly proteins (MRJPs), the neuropeptide corazonin, and venom components) and the *plasma membrane* (mostly olfaction-related) were similarly enriched among the pheromone-sensitive genes in 3/4 species.

There was also cross-species conservation of several GO and KEGG terms related to fatty acid and amino acid biosynthesis

(particularly synthesis of very-long-chain fatty acids, and unsaturated fatty acids, both of which are used in the synthesis of queen pheromone components), lipid transport (including *vitellogenin*), and the KEGG term *Neuroactive ligand-receptor intearctions*. Genes with the molecular function *sequence-specific DNA binding* (e.g. transcription factors) were also pheromone-sensitive.

**Effects of queen pheromone on alternative splicing.** Roughly 20% of genes had ≥2 detectable isoforms, in all four species (Supplementary Fig. 4), allowing us to test for pheromone-sensitive splicing. Pheromone treatment significantly elevated the expression of one isoform and significantly repressed expression of another isoform for 52 genes in *A. mellifera*, 55 genes in *L. niger*, 22 genes in *L. flavus*, and no genes in *B. terrestris* (Fig. 1; Supplementary Tables 11–13). Three genes showed pheromone-sensitive splicing in more than one species, corresponding to around 4% of the maximum numbers of genes that could have overlapped (Fig. 1). Again, these numbers could be underestimates, since we have limited power to detect differential isoform expression, and each 'hit' requires two isoforms per gene to be statistically significant (i.e. we need four significant results to detect a single overlap between species). *DNA methyltransferase 3* showed significantly pheromone-sensitive splicing in *L. niger*, recalling our previous qPCR-based result that queen pheromone affects DNA methyltransferase expression[14]. We next ranked all the alternatively-spliced genes with detectable orthologs in all four species in order of the sensitivity

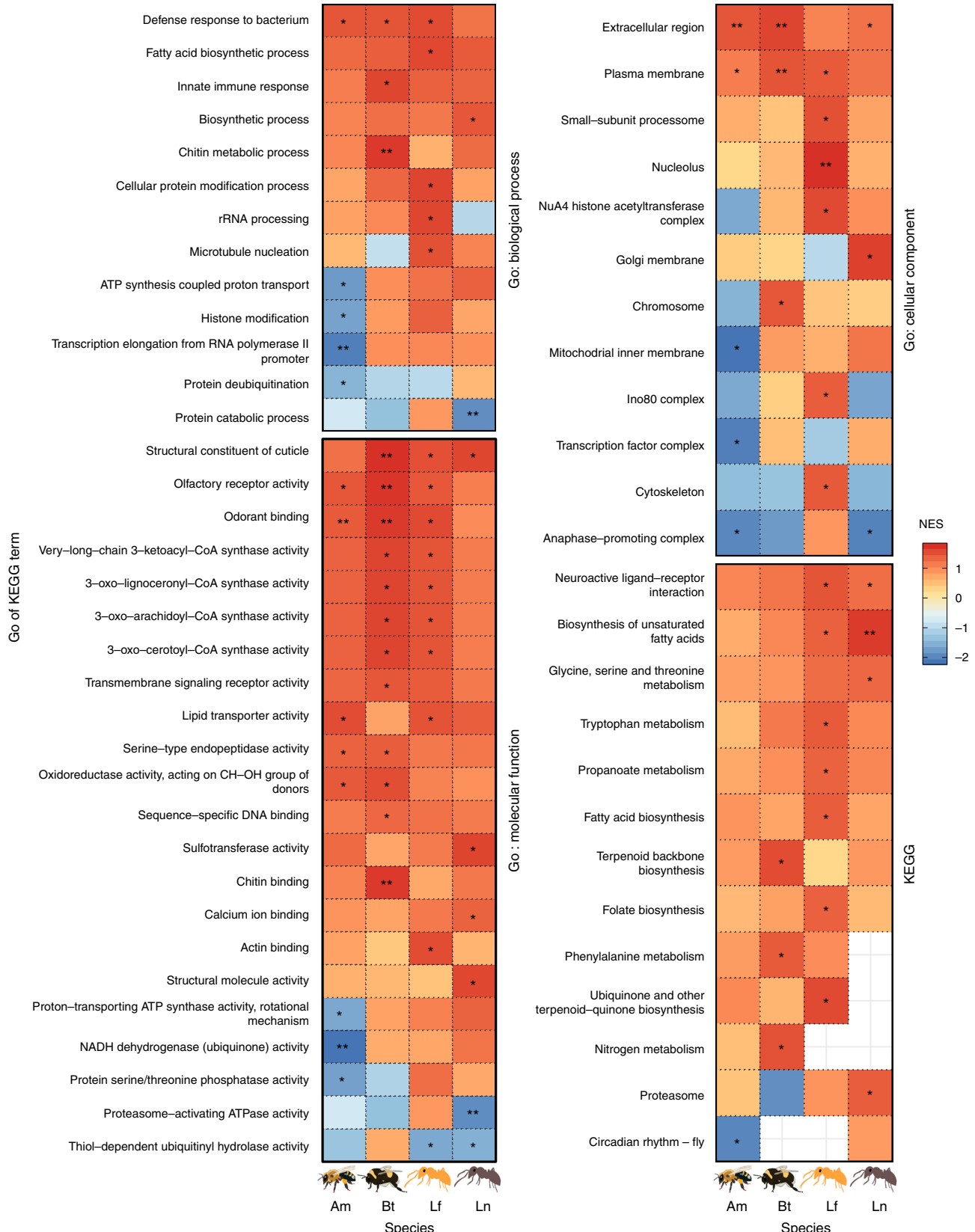

**Fig. 2** Results of Gene Set Enrichment Analysis. The plot shows all Gene Ontology (GO) and Kyoto Encyclopedia of Genes and Genomes (KEGG) terms that were significantly overrepresented (red) or underrepresented (blue) among pheromone-sensitive genes in at least one of the four species. The colour shows the normalised expression score from gene set enrichment analysis. Asterisks denote statistically significant enrichment ($p < 0.05$), and double asterisks mark results that remained significant after Benjamini-Hochberg correction. Empty squares denote cases where we were unable to measure expression for at least five genes annotated with the focal term in one of the species

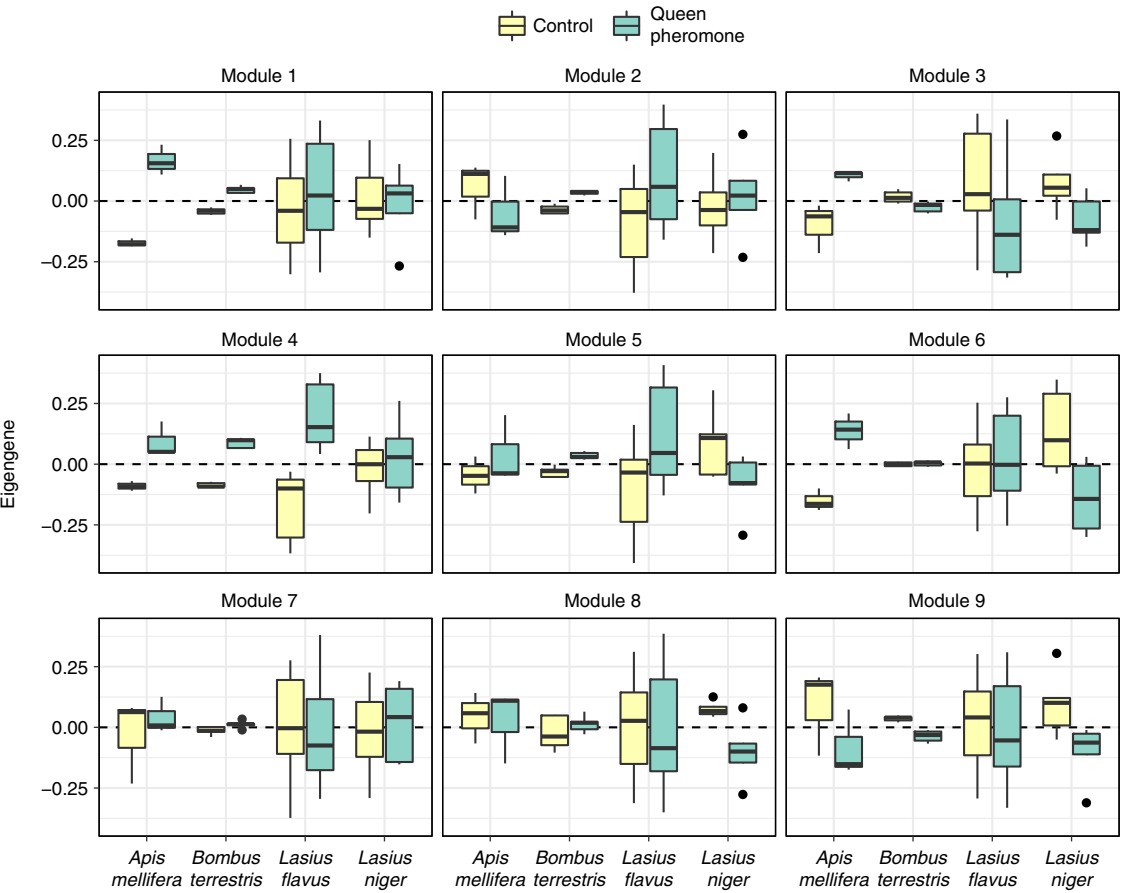

**Fig. 3** Boxplots showing the distribution of 'eigengenes' across samples for each of the nine transcriptional modules identified by WGCNA. The eigengene is a measure of the expression level of a transcriptional module, relative to other samples in the set. Queen pheromone treatment had a statistically significant effect across species for modules 1, 4 and 9 (the annotations give Cohen's d effect size and its 95% CIs, estimated from a multivariate Bayesian model of all nine modules). The boxes show the median and interquartile range, and the whiskers mark the furthest data point within 1.5× the interquartile range of the edge of the boxes (green: queen pheromone treatment, yellow: control)

of their isoform profile to pheromone treatment (defined as the range in log fold change of the focal gene's isoforms), and performed gene set enrichment analysis on the resulting 'splicing sensitivity score'. Most GO and KEGG terms were enriched in similar patterns across species, e.g. *intracellular signal transduction, transmembrane transport, transcription, DNA-templated,* and *serine-type endopeptidase activity* (Supplementary Fig. 5; Supplementary Table 23). The GO terms showing non-significant trends toward enrichment included signal transduction, methyl-transferase activity, mRNA processing, protein transport and modification, and microtubule motor activity. There was also a positive correlation between species in our measure of the sensitivity of splicing to pheromone treatment for all six possible species pairs, though only 2/6 of these correlations were significant (Supplementary Table 22). The correlation was esepcially strong for the two bees ($\rho = 0.19$, FDR-corrected $p < 10^{-7}$), and was also signficant for *Bombus* and *Lasius flavus* ($\rho = 0.09$, FDR-corrected $p = 0.030$). These results suggest that queen pheromone affects the splicing of some of the same orthologs across species.

**Pheromone-sensitive genes tend to pre-date the split between ants and bees**. In all four species, the average pheromone sensitivity (i.e. absolute log fold difference between treatments) of ancient genes (defined as those with a detectable ortholog in ants and as well as bees) was approximately double that of genes that

are putatively specific to bees or ants (Mann–Whitney tests, $p < 10^{-15}$). This result suggests that most pheromone-sensitive genes existed prior to the evolutionary divergence of bees and ants, and thus pre-date the origin of eusociality.

**Effects of queen pheromone on the gene coexpression network**. Among the 3465 genes for which orthologs were detected in all four species, we identified nine modules of coexpressed genes, each containing between 38 and 1639 genes; 3% genes were left unassigned to a module (Figs. 1c and 3; Supplementary Tables 25–34). The best-fitting multivariate model of the nine modules' 'eigengenes' (a metric that quantifies the relative expression of entire modules; see Methods) contained Treatment as a predictor, but not Species or the Treatment × Species interaction (posterior model probability >99%; see Supplementary Table 14). This result suggests that some modules of coexpressed genes responded to pheromone treatment, and that the response is consistent across species. Specifically, modules 1, 4 and 9 showed a statistically significant difference in mean eigengenes between pheromone treatments (Figs. 1c and 3; Supplementary Table 15).

The pheromone-sensitive module 1 was large (1639 genes), and was enriched for GO and KEGG terms related to the cell cycle, DNA repair, transcription and splicing of RNA, and ribosomes (Fig. 4 and S7-S9; Supplementary Table 26). Module 1 also contained genes relating to the epigenome, such as *DNA*

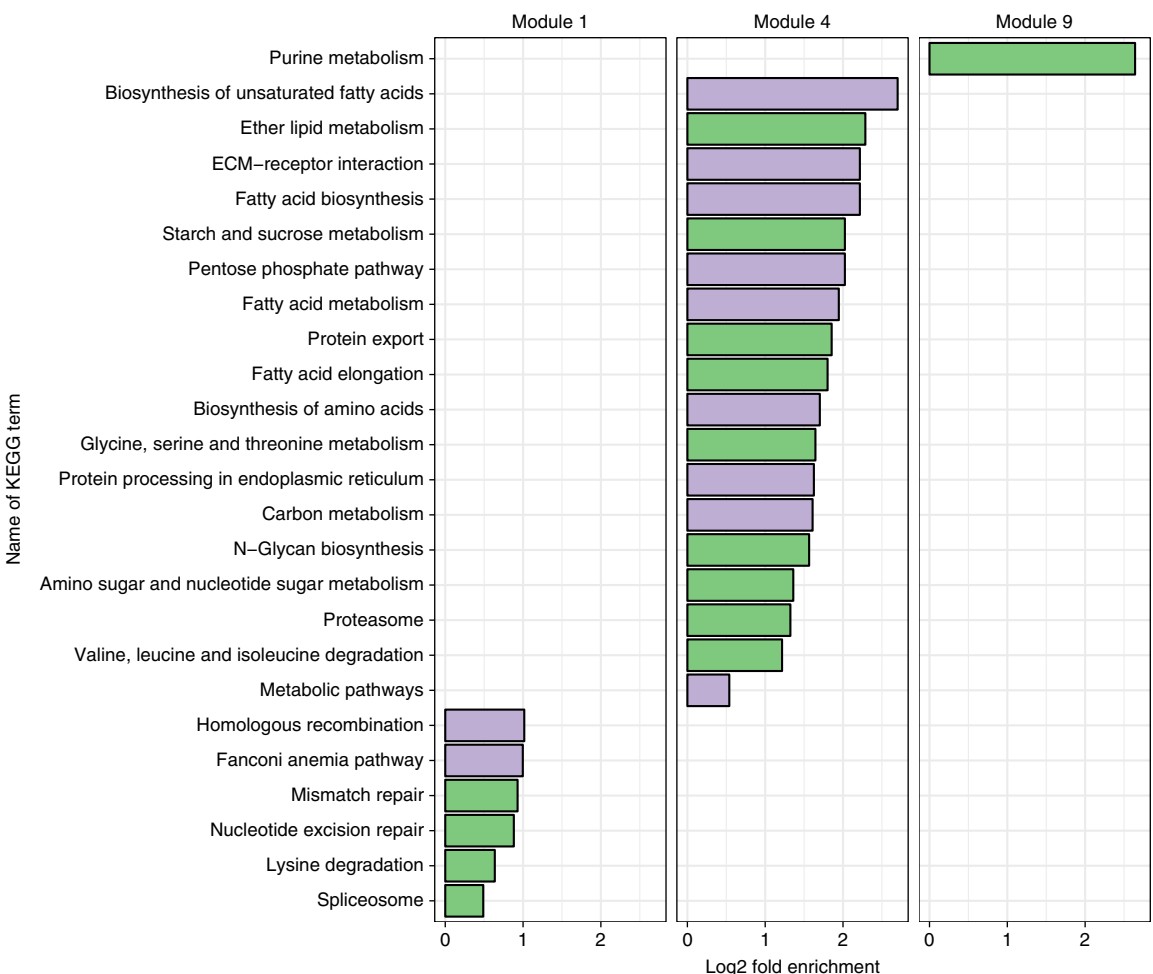

**Fig. 4** Results of KEGG pathway enrichment analysis for the genes in each of the three significantly pheromone-sensitive transcriptional modules. The gene universe was defined as all genes for which we found an ortholog in all four species (i.e. the set that was used to discover these co-expressed modules). All KEGG terms shown in green were significantly enriched ($p < 0.05$), and those shown in purple remained significant after correction for multiple testing. Fold enrichment was calculated as the proportion of genes associated with the focal KEGG term in the module, divided by the equivalent proportion in the gene universe. Supplementary Figs. 6–8 show equivalent results for GO terms

*methyltransferase 3* and several histone deacetylases and methyl-transferases. Module 4 (288 genes) was enriched for GO terms relating to the pentose phosphate pathway, fatty acid and amino acid biosynthesis, lipid metabolism, vesicle-mediated transport, and for genes associated with the endoplasmic reticulum (where proteins, lipids, and steroid hormones are made). This module also contained genes for synthesising very long-chained fatty acids and acetyl-CoA, which are precursor substances for the synthesis of CHCs[24,25] and the main components of honeybee queen mandibular pheromone (QMP)[26]. Module 9 was enriched for purine metabolism; purines are required for cell division and transcription, and to produce important biomolecules like ATP, NADH, and coenzyme A.

**Pheromone-sensitive genes have low connectedness**. We found a negative correlation between sensitivity to queen pheromone and connectedness across genes (Spearman's $\rho > 0.24$, $p < 10^{-48}$ for all species). This means that highly pheromone-sensitive genes are expressed comparatively independently of the rest of the transcriptome, while highly connected genes tend to be insensitive to queen pheromone. This result is illustrated by the excess of pheromone-sensitive genes in Module 0 (which holds

the few genes that were expressed relatively independently of Modules 1–9) in Fig. 1c.

**Characteristics of pheromone-sensitive genes in *Apis mellifera*.** Figure 5 summarises the correlations across genes for a number of gene-level properties, for honeybees. On average, strongly pheromone-sensitive genes had less gene body DNA methylation, lower expression levels, and lower codon usage bias. Pheromone-sensitive genes had higher values of $\gamma$, meaning that they have been under stronger positive selection and/or weaker purifying selection[27]. We also found a positive relationship between pheromone sensitivity and the extent to which a gene was upregulated in queens relative to sterile workers (as measured in ref. [28]). We did not find a significant correlation between a gene's pheromone sensitivity and the caste-specificity of its histone modifications (averaged across the gene, using published ChIPseq data[29]). However, almost all variables were strongly inter-correlated (Fig. 5), making the causal relationships among them (if any) difficult to infer without further evidence.

**Comparison with caste-specific gene expression in ants.** Hypergeometric tests revealed six instances in which one of our gene co-expression modules overlapped significantly with one of

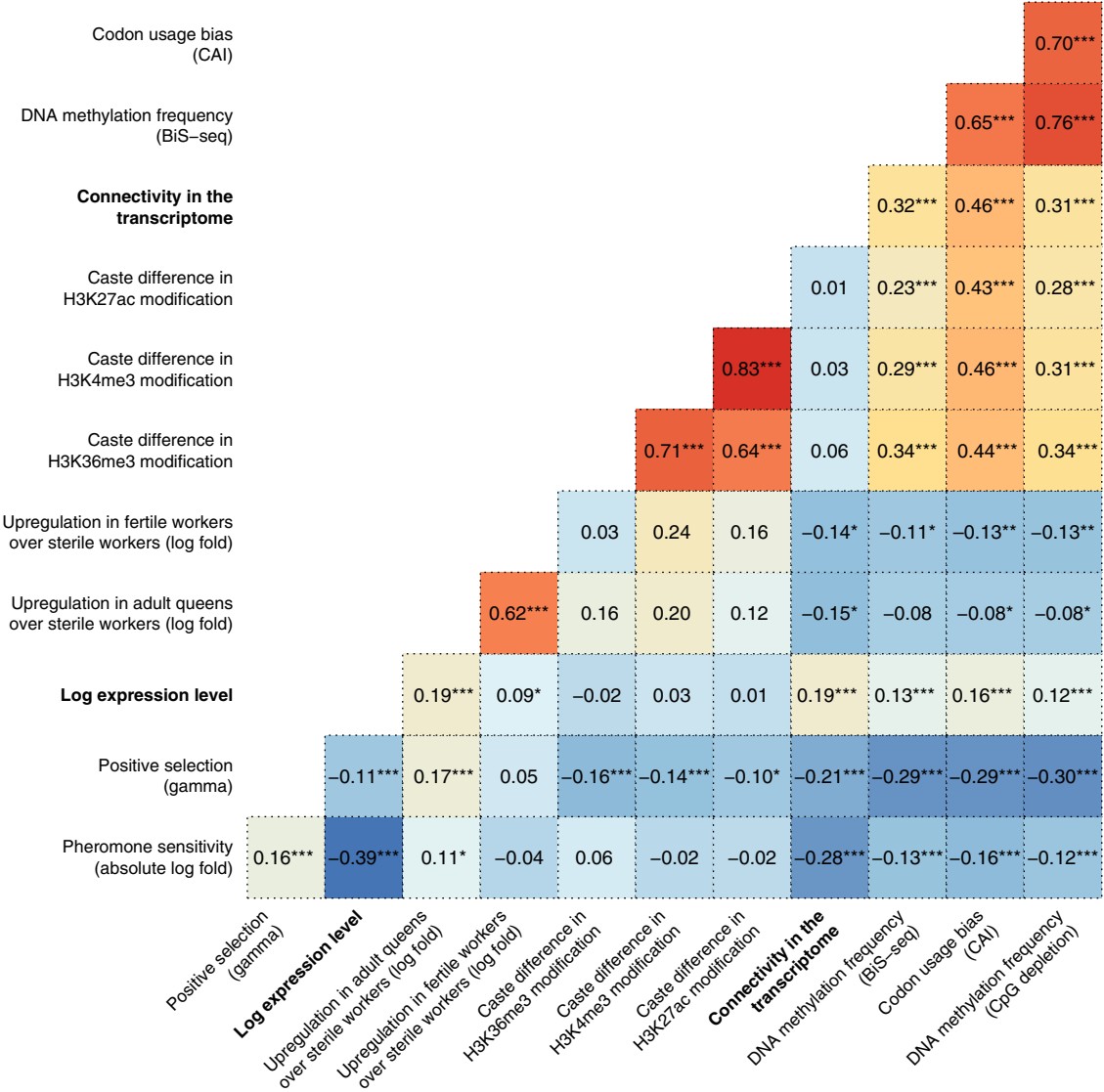

**Fig. 5** Spearman correlations for various gene-level measurements from the present study and earlier research, for *Apis mellifera*. Measurements from the present study are shown in bold: 'Pheromone sensitivity' was calculated as the absolute value of the Log₂ fold difference in expression between pheromone treatment and the control. Expression level shows the logarithm of the average across our six *Apis* libraries. For the 'Upregulation in queens/fertile workers' data[15], positive values denote genes that have higher expression in queens or fertile workers, relative to sterile workers. For the three histone modification variables[29], high values indicate that the modification is more abundant in queen-destined larvae, and low values indicate it is more abundant in worker-destined larvae. The two DNA methylation variables give two different measures of the amount of gene body DNA methylation, namely an indirect measure ($-\log$ CpG O/E ratio) and a direct measure (BiS-seq[80]). Codon usage bias was estimated using the codon adaptation index: high values indicate bias for particular synonymous codons. Lastly, the parameter gamma ($\gamma$) describes the form of selection, where positive values denote positive selection, and negative values purifying selection[27]. The asterisks indicate Benjamini-Hochberg corrected p-values (*$p < 0.05$; **$p < 0.01$; ***$p < 0.0001$)

the modules from Morandin et al.'s study[30] of caste-biased gene expression in ants, after correcting for multiple testing (Supplementary Table 17). Modules 2, 3, and 8 from our study overlapped with worker-biased modules, and Modules 1 and 4 overlapped with queen-biased modules. Since Modules 1 and 4 are pheromone-sensitive (Fig. 3), these results suggest that the set of pheromone-sensitive genes overlaps with the set of caste-biased genes in ants (in addition to honey bees; Fig. 5). Ten genes were found in both our Module 4 and Morandin et al.'s queen-biased module (Supplementary Table 14); these genes included *protein takeout-like, NAD kinase 2, mitochondrial-like, histone H2A-like* and *methyltransferase-like*, again implicating *takeout*, metabolism, and epigenetic processes in caste polyphenism and the response to queen pheromone.

## Discussion
As shown in the figures, some but not all effects of queen pheromones on the transcriptome were similar across the four species. For example, orthologous genes in bees and ants tended to show similar levels of pheromone sensitivity, and we identified three transcriptional modules showing a consistent response to queen pheromone across species. Accordingly, gene set enrichment analysis revealed that broadly similar functional categories of genes were enriched in bees and ants. This cross-taxon similarity is not unexpected, given that queen pheromones induce similar phenotypes (e.g. sterility) in both taxa. However, this outcome was not a foregone conclusion, for example because bees and ants evolved eusociality independently (and have few caste-specific genes in common[31]), and because bumblebees have smaller,

annual colonies in which behavioural interactions play a larger role in regulating reproductive division of labour[32], unlike *Apis* and *Lasius* which have large, perennial colonies with less overt reproductive conflict.

In *Apis* and both *Lasius* species, we found that queen pheromone treatment caused statistically significant changes in alternative splicing at multiple loci, increasing the expression of certain isoforms while inhibiting that of others. The lack of a significant effect in *B. terrestris* might be a false negative, since the estimated sensitivity of alternative splicing to queen pheromone was strongly correlated across pairs of orthologous *Apis* and *Bombus* genes. Also, *Bombus* genes with high (though non-significant) pheromone-sensitive splicing were significantly enriched for similar GO and KEGG terms as those in the other three species (Supplementary Fig. 5). Our study thus adds to the growing list of cases in which alternative splicing underlies polyphenisms in insects[20,33–35].

In *Apis*, pheromone-sensitive genes tended to be positively selected, weakly connected in the transcriptional network, weakly expressed, and hypomethylated, relative to pheromone-insensitive genes. Additionally, queen pheromones affected a somewhat similar set of genes to that which distinguishes adult queens and workers in bees as well as ants, consistent with our prediction that queen pheromones would make gene expression more worker-like. Finally, pheromone-sensitive genes were disproportionately likely to pre-date the divergence of bees and ants, consistent with their patterns of enrichment for GO and KEGG terms associated with taxonomically ubiquitous processes such as lipid biosynthesis.

Many genes or gene families were differentially expressed in two or more species. As one example, queen pheromone inhibited the expression of MRJPs in honeybees (echoing earlier findings that MRJP expression covaries with reproductive physiology[36]) and in *L. niger*. Among other functions, MRJPs are essential for rearing new queens, which workers do when their current queen dies, leaves, or becomes infertile (i.e. when her pheromone disappears)[7]. MRJPs are produced during development and in the adult fat body, and belong to the phylogenetically ancient *yellow* gene family[37], which has diverse roles in development, the nervous system, behaviour, immunity, and pigmentation.

Genes related to synthesis and transport of lipids and fatty acids formed a strongly co-expressed transcriptional module, which was modulated by queen pheromone across taxa. The affected genes included enzymes for making long-chained fatty acids and fatty acyl-CoAs, which are biosynthetic precursors of CHCs as well as the components of the QMP of honeybees[24–26]. Additionally, a number of genes putatively involved in CHC and QMP biosynthesis, such as cytochrome P450s, NADPH synthases, and genes involved in fatty acid elongation and oxidoreductase activity[24–26,38], were differentially expressed. We also observed large (though non-significant) effects of queen pheromone on the expression of *vitellogenin* (a lipid transporter) and *hexamerin 70a precursor*, two classic 'eusociality genes' that have been linked to caste and oogenesis by many previous studies (e.g. refs. [39,40]). These results are expected given that pheromone-deprived workers begin depositing yolk in their ovaries via lipid synthesis and transport[41].

Our results hint at the mechanism by which queen pheromones achieve their effects, and suggest a novel (and heretofore missing[11,12]) mechanism underlying the widely-observed honest signalling of fecundity via olfactory cues/signals in social insects. This honest signalling is considered a puzzle because of the apparent fitness benefits of exaggerating one's fecundity via pheromones (in queens) or of 'covert' reproduction without pheromonal signalling (in workers)[1,5,11,12,42]. We speculate that the fatty acid-derived queen pheromones found in ants and bees

are absorbed directly into workers' bodies (e.g. by ingestion), where they inhibit lipid biosynthesis via negative feedback, thereby inhibiting oogenesis. If this hypothesis proves correct, the colony could be regarded as having a shared 'social physiology', whereby colony members keep track of their own physiological state via standard within-body signals (e.g. juvenile hormone, insulin signalling), as well of the states of other individuals via pheromones[43]. We also speculate that workers evolved elevated sensitivity to queen pheromones as colonies grew larger over evolutionary time, e.g. via changes in behaviour, olfaction, and physiology, allowing them to continue to express the sterility response as their average proximity to the queen declined. Lastly, the necessity of lipid synthesis and transport for oogenesis, coupled with an inextricable, non-evolvable link between the makeup of the internal and external lipid profiles, would enforce a reliable correlation between individual fecundity and odour profile[5,11].

In another notable result, the gene *protein takeout-like* was among the most strongly pheromone-sensitive genes in all four species. The *takeout* family encodes proteins that are expressed in, and secreted from, the brain-associated fat body and antennae, and some members putatively bind juvenile hormone[44]. Interestingly, *takeout* genes have been linked to discrete polyphenisms in termites[45], locusts[46] and aphids[47], suggesting that they might be similarly important to reproductive division of labour in the eusocial Hymenoptera. Additionally, in *Drosophila*, the expression of *takeout* is stimulated by the male-typical isoforms of the sex differentiation genes *doublesex* and *fruitless*, and suppressed by the female isoforms[48]. This is noteworthy in light of the recently-proposed hypothesis that sex differentiation genes such as *doublesex* have been co-opted to control caste polyphenism in eusocial insects[18–20]. This finding also brings to mind the reproductive groundplan hypothesis[43], which posits that reproductive division of labour is the result of regulatory evolution of nutrition signalling pathways, e.g. insulin-like signalling, which (among other things) controls the balance of lipid and sugar synthesis and metabolism, and the rate of ageing (which differs between queens and workers, and between fertile and sterile workers[22]).

Additionally, *serotonin receptor* was among the most pheromone-sensitive genes in all four species. Serotonin seems understudied in social insects, although two studies have found differences in serotonin titre or receptor gene expression between sterile and fertile workers, in *Polistes* wasps[49] and *Apis*[50]. Another biogenic amine—dopamine—is better-studied; it has been implicated in the response to queen pheromone in *Apis*[51], and affects behaviour and fecundity in many insects[52–54]. There was also some evidence that the expression of the neurohormone corazonin was modulated by queen pheromone (e.g. Supplementary Table 18), consistent with experimental results showing that corazonin induces worker-like behaviours and suppresses queen-like behaviors in an ant[55].

Several genes related to myosin, which functions in muscle contraction, were significantly downregulated in the queen pheromone treatment in at least one species. Interestingly, a recent study compared the transcriptomes of queens and workers in 16 ant species with RNA-seq, and found only a single gene that was significantly differentially expressed between castes in all 16 species: another myosin gene[30]. Myosin genes are also differentially expressed between fertile and sterile workers[56,57] and queen- and worker- destined larvae[58] in honeybees, and between queens and workers in bumblebees[21]. We speculate that myosins show caste-specific expression due to caste differences in muscle morphology and activity levels (e.g. queen ants fly while workers do not, and in bees, workers fly more than queens).

The pheromone-sensitive module 1 contained many genes relating to histone modification, particularly histone-lysine

N-methyltransferases and histone deacetylases. These included orthologs of *histone-lysine N-methyltransferase eggless*, which trimethylates Lys-9 of histone H3 in the *Drosophila* ovary, and which is essential for oogenesis (FlyBase: FBgn0086908). Another interesting gene was *male-specific lethal 1 homolog*, which regulates gene expression by acetylation of H4 lysine 16; the resulting H4K16ac 'epimark' is hypothesised to regulate the development and renewal of female germline stem cells[59]. Another histone acetylation epimark, H3K27ac, is related to the major-minor worker size polymorphism in carpenter ants[60], and differs between queens and workers in honeybees[29]. In sum, it seems likely that the receipt of queen pheromone causes a rewiring of the epigenome, which in turn regulates the genes underlying oogenesis. We also found that queen pheromone affected the splicing of *DNA methyltransferase 3* (*dnmt3*) in *Lasius niger*, echoing our previous work showing that queen pheromones affect the expression of *dnmt1* and *dnmt3* in bees and ants[14], and paralleling evidence that differential DNA methylation is involved in queen-worker polyphenism[61]. Direct measurement of the effect of queen pheromone on the epigenome has not yet been performed, to our knowledge.

A number of recent papers on the origins of eusociality have asked whether the key genetic players tend to be ancient genes with fundamental cellular functions, or more recently-evolved genes with specialised functions[30,31]. Most of this work has focused on genes showing queen- or worker-biased expression, but since that gene set overlaps substantially with the set of pheromone-sensitive genes, our results are apposite. We found that pheromone-sensitive genes tend to predate the split between bees and ants, suggesting that present-day queen pheromones primarily affect genes that already existed in the genomes of the first eusocial insects. However, we also found that pheromone-sensitive genes had low connectedness, expression levels, and codon usage bias; none of these characteristics are consistent with the targets of queen pheromone being 'housekeeping' genes, i.e. extremely ancient, constitutively-expressed genes with ubiquitous cellular functions[30]. Instead, queen pheromone affected a moderately-sized subset of the transcriptome, whose expression varied relatively independently of the remainder. This result is interesting because genetic modules showing flexible expression patterns and reduced pleiotropy are predicted to be major drivers of adaptation because they are comparatively free to undergo adaptation[18]. Our results are consistent with a model whereby a relatively self-contained genetic module (controlling nutrient homoeostasis, and thus oogenesis) acquired a new expression pattern, producing the observed polyphenism of fertile and sterile females. Subsequently, the genes in this module underwent adaptation to their new roles, explaining our result that pheromone-sensitive genes are both evolutionarily ancient and positively selected.

## Methods

**RNA sequencing of pheromone-treated bees and ants.** The present study uses RNA obtained from the same insect samples as those used in an earlier study[14], which provides complete methods for the pheromone bioassay, RNA extraction, and preparation of cDNA. Briefly, we treated nest boxes containing workers from 3 to 8 colonies per species with either a solvent-only control or their own species' queen pheromone, and then extracted total RNA from individual workers (either whole bodies, or a random lateral body half for *Bombus*), removed genomic DNA with DNase, and reverse-transcribed RNA to cDNA. For *Apis mellifera* honeybees, the pheromone used was commercially available QMP, which is a mixture of five chemicals (principally keto acids). For *Bombus terrestris*, the pheromone was the CHC pentacosane ($C_{25}$), and for the two *Lasius* ant species it was the CHC 3-methylhentriacontane (3-MeC$_{31}$). The *B. terrestris* workers were from the same cohort and colonies as in Holman[62], though they were different individuals.

We then used Qubit fluorometry to determine the mass of cDNA obtained from each worker, and pooled equal amounts of cDNA from five randomly-selected workers for each combination of species, colony, and treatment. Not including four problematic samples that were later discarded (see Supplementary Figs. 1–2), we

sequenced 44 cDNA pools (6 for *A. mellifera*, 10 for *B. terrestris*, 13 for *L. flavus*, and 10 for *L. niger*; Supplementary Table 1); library preparation (using Illumina TruSeq kits) and sequencing was conducted by Edinburgh Genomics. The libraries were sequenced in three lanes of an Illumina HiSeq 2500 sequencer set to high output, yielding 125 bp paired-end reads. All samples were individually barcoded and run in all three lanes, preventing lane effects from confounding the experiment. The experiment yielded $14 \pm 1.3$ (st. dev.), $12.3 \pm 4.3$, $14.7 \pm 2.7$, $13.1 \pm 1.4$ million reads for *A. mellifera*, *B. terrestris*, *L. flavus* and *L. niger*, respectively. We used Trimmomatic[63] to remove sequencing primers, and trimmed reads for quality using the SLIDINGWINDOW:4:15 parameter prior to subsequent analyses. After trimming, the number of paired reads was $10.8 \pm 1$, $8.3 \pm 2.9$, $10.2 \pm 1.4$, $10.8 \pm 1.3$ million, respectively.

**Quantifying differences in gene expression or alternative splicing between pheromone treatments.** We aligned and quantified the raw reads using the RSEM[64] pipeline with Bowtie2[65] to transcripts from published genomes for *A. mellifera*, *B. terrestris* and *L. niger*[66–68]. The *L. niger* genome assembly had no isoform information, so we identified isoforms using Tophat[69]. No reference genome was available for *L. flavus*, so we assembled the transcriptome de novo using Trinity[70], and identified coding regions with TransDecoder[71].

Within each species, we used EBSeq-HMM[72] to calculate the fold difference in expression between the control and pheromone-treated workers for each transcript using the rsem-run-ebseq pipeline implemented in Trinity. We adjusted p-values to control the false discovery rate using the Benjamini-Hochberg method, then defined genes with adjusted $p < 0.05$ as significantly differentially expressed. As a sensitivity analysis, we re-ran the EBSeq analysis after removing low-abundance transcripts, increasing power by reducing the number of tests; we obtained essentially identical lists of significant genes to those from the full analysis.

To identify genes whose splicing was significantly affected by queen pheromone, we searched for genes for which at least one isoform was significantly upregulated in the control, while another isoform was significantly downregulated. We also calculated a 'pheromone sensitivity of splicing' score, by taking the maximum difference in log fold change for the isoforms of each gene (e.g. a gene with three isoforms showing $-2$, $+0.1$, and $+1$ log fold difference between treatments would score 3). This score was use to test for correlations across species in the gene-level sensitivity of splicing to pheromone, and for GO and KEGG enrichment tests of pheromone-sensitive splicing (see below).

**Testing for conserved effects of queen pheromone across species.** The simplest method to identify conserved effects on gene expression is to tally the number of orthologous genes showing significant differential expression in two or more species (as in the Venn diagrams in Fig. 1). Though robust, this method is highly conservative, because one has finite statistical power to detect any given differentially expressed gene. Power issues are compounded when searching for genes that show a conserved response across species, since one needs to avoid two or more false negatives per locus. We therefore performed two additional formal analyses to test for conserved transcriptional effects of queen pheromones, as well as plotting the pheromone sensitivity for each gene (Fig. 1c) to allow qualitative assessment of the extent of cross-species similarity.

For the first formal test, we tested whether the pheromone sensitivity of each gene is correlated in each pair of species, using Spearman's rank correlation on pairs of orthologous genes (see Fig. 1c, d). Pheromone sensitivity was defined as the absolute log fold difference in expression between treatments. This test has improved power relative to the Venn diagram approach, but does not reveal the number or identity of the conserved/convergent pheromone-sensitive genes.

Secondly, we identified the genes that had detectable orthologs in all four species (defining orthologs as genes that were each other's best BLAST hit, with both e-values $<10^{-4}$), and then ranked them from most to least pheromone-sensitive within each species. Then, we asked which genes appeared in the top *n*-most pheromone sensitive genes in all four species, for $n = 100, 200… 500$. This analysis has good power to identify candidate genes that responded to pheromones in all four species, but runs the risk of false positives (i.e. genes that topped all four gene lists by chance alone).

**Evolutionary age of pheromone-sensitive genes.** To test whether pheromone-sensitive genes tend to have an ancient or recent evolutionarily origin, we classified genes as either 'ancient' or 'putatively family-specific' using reciprocal best BLAST. Bee genes (*Apis* or *Bombus*) with a BLAST hit (e-value $10^{-4}$) in at least one of the ant species were classified as ancient, and vice versa. Genes that were not classified as ancient might be false negatives (e.g. due to gaps in our sequence data, or because genes were lost in one lineage), hence our caution in labelling genes as family-specific. Any misclassifications should make it harder to detect a difference between pheromone-sensitive and -insensitive genes, but could not produce a spurious difference.

**Gene co-expression network analysis.** We constructed a gene co-expression network for all four species, which included all genes for which orthologs were detected in all species, following Morandin et al.[30]. The aim of this analysis was to search for 'modules' of co-expressed genes that change their expression in response

to queen pheromone in all the species. We therefore used an empirical Bayes method[73] (implemented via the ComBat function in R's *sva* package[74]) to transform the expression data so as to remove multivariate differences in expression attributable to species or colony, clarifying the effect of pheromone treatment on the transcriptome.

We used the R package WGCNA[75] to define the gene co-expression network and identify transcriptional modules, largely using the default settings. The two exceptions were that we imposed a minimum size for transcriptional modules of 30 genes, and used a signed (rather than unsigned) coexpression network. These choices mean that our analysis recovers modules of 30+ genes that are all simultaneously up-regulated or down-regulated across our 44 samples.

To test whether species, treatment, and their interaction explained variation in module 'eigengenes' (a metric describing the expression level of a particular module in the focal sample, relative to the other samples[75]), we used Bayesian multivariate models implemented in the R package brms[76]. We fit five candidate models, all with the nine eigengenes as a multivariate response, colony as a random effect, and Gaussian errors. The five models differed in their fixed effects, and we compared the models' fits in order to test for significant effects of treatment, species, and their interaction (using posterior model probabilities, calculated using bridge sampling).

We also used the co-expression network to calculate connectedness for all genes. We defined connectedness as the sum of the correlations in expression between a given gene and every other gene in the network[77]. Thus, a 'highly-connected gene' is one whose expression varies in concert with many other genes, across samples.

**GO and KEGG enrichment analyses**. We downloaded KEGG from the KEGG API and GO terms from NCBI, for the best-annotated of our four species, *Apis mellifera*. KEGG terms group together genes that are known to interact in biochemical pathways, while GO classifies genes by Biological Process, Molecular Function, or Cellular Component. Genes in non-*Apis* species were assumed to have the same GO and KEGG terms as their reciprocal best BLAST hits in *A. mellifera*.

We implemented Kolmogorov-Smirnov enrichment tests (also called Gene Set Enrichment Analysis or GSEA[78]) using the fgsea package for R. These tests rank all genes in the set under test (called the 'gene universe') in order of some metric of interest, and then identify GO or KEGG terms that are significantly over-represented and under-represented among the top-ranked genes, relative to random expectations derived by bootstrapping. As well as presenting the uncorrected *p*-values, we corrected the GO and KEGG results for multiple testing using the Benjamini-Hochberg method, though we note that this approach is crude and probably overly-conservative, since tests of the different terms are not independent. GO results were simplified by collapsing redundant GO terms into higher-order ones using the *collapsePathways* function in *fgsea*.

To identify enriched GO and KEGG terms among genes whose expression was sensitive to pheromone treatment, we ranked genes by the $\log_{10}$ posterior probability of differential expression (computed by EBSeq-HMM) and defined the gene universe as all genes (for *Apis*) or all genes with a detectable ortholog in *Apis* (for other species). To identify enriched terms among genes with pheromone-sensitive splicing, we ranked genes by their splicing score, and specified the gene universe as all alternatively-spliced genes with *Apis* orthologs.

To identify enriched GO and KEGG terms for each of the ninico-expressed genetic modules, we used standard hypergeometric tests (implemented in the *clusterProfiler R* package), and defined the gene universe as all 3465 genes used in the coexpression network analysis.

**Characteristics of pheromone-sensitive genes in Apis**. *Apis mellifera* honeybees are well-studied relative to our other species, and so we compared our pheromone sensitivity and connectedness data to pre-existing gene-level data from *A. mellifera* using Spearman correlations.

First, we used published microarray results[28] to test whether pheromone-sensitive genes also showed a large difference in expression between A) queens and sterile workers, and B) fertile workers and sterile workers. Second, we examined codon usage bias, as measured by the codon adaptation index[79]; high values indicate a bias towards particular synonymous codons in the coding regions of a gene. Third, we tested for relationships with the frequency of DNA methylation within the gene body, using two complementary measures of DNA methylation: the amount of CpG depletion (measured as the negative log observed/expected CpG ratio), or the percentage of methylated cytosines, estimated using whole genome bisulphite sequencing[80]. Fourth, we tested whether pheromone-sensitive genes show signatures of positive or purifying selection since the split between *A. mellifera* and its congeneric *A. cerana*, using the metric $\gamma$[27]. Lastly, we tested whether pheromone-sensitivity was correlated with the log expression level of each gene, using the average expression levels from the present study.

**Comparison with caste-specific gene expression in ants**. Using reciprocal best BLAST (*e*-value $10^{-4}$), we attempted to classify the groups of orthologous genes from our study into one of the orthologous gene groups defined for queens and workers from 16 ant species in Morandin et al.[30]. We then tested for significant overlap between that study's 36 gene co-expression modules, and the modules from our own study, using hypergeometric tests on all possible pairs of modules

(followed by FDR correction). We thereby tested whether the groups of coexpressed genes that respond to pheromone also tend to show differential expression between queens and workers.

**Reporting Summary**. Further information on experimental design is available in the Nature Research Reporting Summary linked to this article.

## Data availability

The raw sequencing data have been deposited at NCBI (BioSample ascensions: SAMD00106316 to −58 [https://www.ncbi.nlm.nih.gov/bioproject?LinkName=biosample_bioproject&from_uid=10236766]). All the remaining data used to generate the results in this paper are available at https://github.com/lukeholman/queen-pheromone-RNAseq and https://osf.io/s6frk/.

## Code availability

Bash, Python, and R scripts used to reproduce our bioinformatics pipeline and data analysis are archived on Github (https://github.com/lukeholman/queen-pheromone-RNAseq). An HTML report showing all the code used for bioinformatics and to generate all our results, statistics, and plots can be viewed online at https://lukeholman.github.io/queen-pheromone-RNAseq/. A timestamped version of the code is archived at https://osf.io/s6frk/.

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

## Acknowledgements
We are very grateful to Lijun Qiu, Tapio Envall, Sini Vuorensyrjä, Heini Ali-Kovero and Minna Tuominen for laboratory assistance, to Soojin Yi and Brendan Hunt for sharing data, and to Mark and Kirsten Holman for beekeeping. Brian Hanley and Jocelyn Millar kindly provided synthetic 3-MeC$_{31}$. Claire Morandin and Tim Linksvayer provided helpful discussion. This work received funding from the Research School of Biology at Australian National University to LH; a Discovery Project (DP170100772) to LH and ASM; the Kone Foundation to HH; the Academy of Finland to HH (135970, 127390), and the Center of Excellence in Biological Interactions (284666).

## Author contributions
L.H. designed and managed the project, conducted the bee experiments, performed bioinformatic and statistical analyses, and wrote the manuscript. H.H. conducted the ant experiments and co-wrote the manuscript. K.T. carried out lab work and co-wrote the manuscript. A.S.M. performed bioinformatic and statistical analyses and co-wrote the manuscript.

## Additional information

**Competing interests:** The authors declare no competing interests.

