## [Peer Review File · Nature Communications]

Reviewers' Comments:

Reviewer #1:

Remarks to the Author:

In this study, the authors conducted transcriptional profiling of workers in 4 different species (2 bees and 2 ants) with and without exposure to queen pheromones. Their analysis revealed many genes with differential expression or alternative splicing. Some of these genes, gene categories (GO and KEGG), or gene modules (groups of co-expressed genes) displayed similar direction of expression change across taxa, possibly indicative of "conserved effects on the transcriptome". They also found that pheromone sensitive genes typically predate the divergence of ants and bees, have low connectedness, and often share particular characteristics (low gene body methylation, low expression levels, low codon bias, etc) in the honeybee. They also compared their gene modules with those from a previous study examining workers versus queens.

This study has generated a lot of data and the analysis appears sound. This study will be the basis for many useful follow up studies.

While I really liked the idea of this study, I am not fully convinced that the main conclusion or interpretation (i.e., conserved effects on the transcriptome) is well supported by the results.

1) The authors studied 2 ants (same genus) and 2 bees (the honeybee and the bumble bee) but the bumblebee showed almost no differences with pheromone treatment (1 gene for differential expression, Line 71; 0 genes for splicing, Line 81; Fig 1 left panel). One possibility is that there was not enough power (perhaps unknown factor associated with sample size, sub-optimal stage, pheromone dose/potency, etc) to detect the differences. In some ways, this is partially addressed in Fig 1 (right panel) by correlation in pheromone sensitivity across species. However, another possibility is that they don't really share the gene expression patterns, which appears to be the opposite of their main message.

2) Related, the overlap in gene expression or splicing is not different from random when comparing the honeybee (*Apis mellifera*) to either of the ants (Fig 1 Left panel; lines 97-99). To me, this does not seem to support conservation of a queen pheromone effect, and may indicate "neutral" patterns. Similarly, for the KEGG pathway analysis (Fig 2), it seems only the "ATP hydrolysis coupled proton transport" is up regulated in the controls for the honeybee plus the 2 ants (and no bumble bee, as indicated above). For GO (Fig 3), there are 2 categories (carbon metabolism and phototransduction in the fly). Note, there was much more overlap between the 2 ants, but this is expected because they are in the same genus.

Other minor comments:

1) Figure 3 legend. "... for which at least two of the nine enrichment tests returned a significant result..." Should this be "one of the nine"? I may have missed something, but it seems some of the categories only have 1 significant results (e.g., RNA transport).

2) Lines 219 to 242. Here the authors discuss several genes which were somewhat highly ranked in their gene lists, but were not significant. And while I realize significance does not always need to meet a hard threshold, this study is based all on bioinformatics analysis. Perhaps complementary experiments would help?

Reviewer #2:

Remarks to the Author:

Queen pheromones have conserved effects on the transcriptome in ants and bees

Summary

This study tests whether chemically similar compounds in social insect queen pheromones elicit similar effects on the transcriptome of different social insect taxa - namely, two species of ant and two species of bee. Specifically, they use RNA-Seq data from a previous study (Holman et al 2016) to identify transcripts differentially expressed upon exposure to species-appropriate queen pheromone. From these data they test the idea that the response to pheromone ought to be 'similar' between species, and in the end draw this conclusion. Their main result is summarized in Figure 1, which they use to argue that i- all four species up-regulate 'similar' genes in response to queen pheromone (Fig 1 - top), ii- these genes are 'similarly' prone to splicing (Fig 1 - bottom), and iii- that these genes tend to show 'similar' pheromone sensitivity (Fig 1 - right). The authors bolden this narrative with other analyses, which include 'similar' KEGG (Fig 2) and GO (Fig 3) enrichment terms, evidence for 'similar' co-expressed gene modules across taxa (Fig 4) and, finally, a series of population genetics parameters that include codon-usage bias, positive selection and methylation tags (Fig 5) that they argue is evidence for pheromone-responsive genes being evolutionarily new (i.e., not ancient housekeeping genes) and having evolved in novel ways under (kin) selection. The data and analysis appear to be technically sound, and does include new and interesting results that may be of interest to the insect sociobiology community. As I highlight below, however, the main conclusion, which is conveyed in the title, is not obviously well supported nor, in some cases, are the results entirely novel.

Major comments

Engagement with literature

L47 'To date only one study - in the honey bee *Apis mellifera* - has experimentally measured the effects of queen pheromone exposure on the transcriptome (Grozinger et al 2003)'. This statement, as written, does not appear to be correct. There are at least half a dozen other such studies, particularly from the Oldroyd group and Wenseleers group, and more studies yet from Grozinger/Robinson group. It was common in the microarray era (2003-2010) to examine transcriptional changes in honey bee worker gene expression upon exposure to queen pheromone. In addition, there are studies that did this on a smaller scale, using RT-PCR on a locus by locus basis.

Strong evidence for conclusions

L60 'Our principle aim was to test whether pheromone-sensitive genes, pathways, and transcriptional modules are similar or distinct across species'. This hypothesis of similar versus distinct is a difficult one to test because no doubt some aspects of the transcriptome response will be similar and other aspects will be distinct. Say for example, GO terms are 25% similar across taxa, or alternatively are 75% similar (and thus 25% distinct). In either case, what do you conclude? That they are similar... or that they are distinct? My point is that there is no a priori threshold or expectation to draw a conclusion either way and the author's choice to emphasis 'similar' in this paper is arbitrary. A critic would ask: similar relative to what expectation? Maybe the four taxa chosen are this 'similar' for *any* chemical exposure (or treatment in general) based simply on phylogeny. If so, the chosen conclusion for conserved social evolution may have little to do with social evolution ...just phylogeny. I would therefore argue that in the category of 'strong evidence for conclusions' there is room for interpretation.

Novel results

The single gene with the biggest expression was Major Royal Jelly Protein 3 (89-fold in QMP-). Indeed, five of the 12 top-most differentially expressed genes were MRJPs (MRJP 1, 2, 3 and 4, plus 3-like). The authors may wish to compare this result to a previous study that examined transcriptome-wide changes in worker gene expression upon exposure to pheromone (Thompson et al 2006. *Ins Mol Biol* 15:637-644). In that study, the authors likewise found multiple MRJP transcripts differentially expressed upon exposure to queen pheromone in *Apis mellifera*, including MRJP2 MRJP3 and MRJP4 as reported here (plus MRJP5 and MRJP7).

Minor comments

L42 Authors should comment on what the 'solitary common ancestor of the social Hymenoptera' was or what that ancestral signal was. Was the signal part of a sexual signalling pathway prior to its co-option into reproductive division of labor?

L51 'The chemical similarity of most species queen pheromones, coupled with the fact that queen pheromones influence similar phenotype traits in diverse lineages, leads us to predict...' What are the 1- similar phenotypic traits and 2- diverse lineages that the authors are referring to?

L62 What is meant by 'targets' of queen pheromones, and what are the mechanisms regarding 'how' signals operate that the authors are alluding to?

L65 What is meant by 'shedding light'? Should spell-out a clear hypothesis to test.

Figure 1- Note that in Fig 1 there really isn't much overlap in the Venn diagrams, including no genes for *Bombus*. What proportion of Differentially Expressed Genes do these numbers represent? If there are more than 3000 shared orthologues between species (L130), then co-expressing between $n = 1$ to 5 of them among non-congeners (Fig 1 - top) or co-splicing between 1 to 2 of them (Fig 1 - bottom) does not seem to be very 'similar', as the authors conclude.

Figure 1 - Note too that the single asterisks denotes one case of overlap that was significantly higher than expected by chance (hypergeometric test, $p < 0.0001$), but this overlap is between the two congeners, so the 'chance' is probably confounded by phylogeny.

Figures 2 and 3 present important results, but this type of output is a bit prosaic: a list of KEGG and GO terms.

Figure 4 is perhaps the most interesting and novel result. The authors detect nine modules of co-expressed genes that contain between 38-1639 genes (Fig 4). Two of these modules significant predictors of pheromone treatment across taxa and are further shown to have low connectedness, which suggests they are expressed independently of the rest of the genome. This is an interesting and novel result.

Figure 5- Gene-level measurements for *Apis mellifera*. Authors describe 'up-regulation' as a fold differences between queens and workers... but do the authors examine queens? I may have missed a subtle transition in the analysis, but in general the comparisons are between pheromone-treated and un-treated workers, so the introduction of queens was unexpected. In general, Fig 5 is hard to interpret.

L 177 'Many genes were significantly affected by pheromone in two species, and (only) a single gene

was significant in all three" : One could interpret this sparse pattern of minimal overlap to mean that the pheromone response is NOT particularly similar between species.

L178 - 'Orthologous genes tended to show a similar level of pheromone sensitivity...': Looking at Fig 1 where the correlations are provided (right panel), it doesn't seem that similar to me. Spearman's P is as low as 0.077 between *Lasius* and *Apis*.

Reviewer #3:

Remarks to the Author:

Review of manuscript by Dr Holman and co-workers entitled "Queen pheromones have conserved effects on the transcriptome in ants and bees", NCOMMS-18-15576.

The authors study patterns of differential gene expression in multiple species of social insects subject to treatments of queen pheromone. They find that different species show some similarities in the patterns of differential expression. In addition, genes differentially expressed tend to show certain characteristics.

Overall, I thought this was an interesting study that was well carried out. The manuscript is generally well written and the analyses are extensive and thought provoking. The study adds to our understanding of differential gene expression in social insect taxa.

Although this study was interesting and well conducted, I'm not sure that it would be broadly interesting to a wide audience. In addition, it is not so surprising that a pheromone putatively involved in the same function in different species would show at least some similar effects on gene expression in different taxa. So, although I viewed the study favorably, I was less sure about its general appeal.

Minor Comments

288 The authors apparently directly treated workers with queen pheromone. But it is not clear that workers ever come into direct contact with queen pheromone in natural settings in the way that it is applied in these experiments. That is, workers might detect queen pheromone if it is fed to them through trophallaxis. Or there might be queen pheromone spread in other non-tactile ways. But workers almost surely do not get queen pheromone applied topically as seems to be the case in these experiments (although the methods were unclear on this). So the experimental effects of topical application of queen pheromone as obtained in this study may not be the same as those in natural conditions.

The authors look for and find many characteristics of genes differentially expressed under the pheromone treatments. I think it is certainly good that the authors did these analyses as the results are useful as presented. But I wouldn't take the results too far. For example, the findings that pheromone sensitive genes also tend to be unmethylated and differentially expressed in other contexts is as expected. Previous studies have shown that genes differentially expressed in one context tend to be differentially expressed in other contexts. And differential expression of genes in insects is also correlated with low methylation. So the point is that genes that are differentially expressed, regardless of the context, tend to show many other correlated properties. But it is unlikely the differential expression in the context of pheromones is causal of these results.

Reviewer #4:

Remarks to the Author:

Holman et al. analyse the transcriptomes of two ant and two bee species upon stimulation with a synthetic pheromone. The authors identify sets of differentially expressed genes in each species and analyse genes that tend to have similar responses upon pheromone stimulation in different species. They perform a comprehensive mining of conservation analyses, enriched sets of genes and co-expressed gene modules. They find that genes that are sensitive to pheromone stimulation tend to be present in the common ancestor of bees and ants. These genes are lowly expressed, have low connectedness and are hypomethylated. The authors argue that a module of genes, which was present in the common ancestor, is commonly stimulated in both ants and bees.

The introduction and the scientific questions are insightful. I appreciate that the supplementary material includes the R code to reproduce the supplementary figures, that there is a Github repository for all the code and that the authors are objective with their data. The downstream analyses of the differentially expressed genes are convincing and the conclusions of their analyses are interesting. However, the main conclusion is contradicted by some of their analyses. I provide more details below.

Major points:

1. The RNA-seq data was generated from either whole bodies or lateral body halves. Is pheromone stimulation expected to have effects on the whole body of the animals or only on specific cell-types? I think it would be helpful for the authors to discuss if/how this could affect their analyses and conclusions.
2. The authors highlight in their title and discussion that transcriptomic changes upon pheromone stimulation are shared across species and tend to be associated with genes present in the common ancestor of bees and ants. I believe that this conclusion stems from the co-expression module analysis. However, this statement is contradicted by the data from Fig. 1: Despite some of the gene overlaps being higher than the overlaps expected by chance, the large majority of stimulated genes seem to be species-specific. For example, in Fig 1, *Lasius flavus* and *Lasius niger* share 29 genes, which is less than 10% of the genes stimulated in *L. flavus*. Moreover, the correlations shown in Fig 1 are positive but weak. Altogether, one could also reach the opposite conclusion that most pheromone-associated changes are species-specific.
3. To what extent are the differences between species in the Venn diagram in Fig. 1 due to thresholding effects (including the fact that *Bombus terrestris* has no significant genes)? It would be good to see more exploratory analyses in this respect; for example, it would be helpful to have a scatterplot of the log₂ fold changes (stimulated vs. control) between pairs of species, in which the points that were detected as significant in one species are highlighted in colour. This approach would provide information about genes with consistent fold changes across species.
4. To avoid that thresholding effects drive the conclusions, I would suggest that the authors re-analyse their data from the different species using linear models (for example multifactor models in edgeR or DESeq2, or ANOVA) or similar analyses. For example, a model `~ species + pheromone_condition` would identify pheromone-stimulated across all samples, and a model with an interaction effect `~ species + pheromone_condition + species:pheromone_condition` would identify species-specific effects.
5. In the gene-set enrichment analyses, it's not clear whether the p-values were corrected for multiple testing. Correcting for multiple testing is necessary, otherwise the authors can't draw conclusions from

this analysis.

Minor:

6. The accession number of the raw sequencing data does not exist in NCBI Biosamples.

Response to comments on “Queen pheromones have conserved effects on the transcriptome in ants and bees”

Reviewer 1 comments

In this study, the authors conducted transcriptional profiling of workers in 4 different species (2 bees and 2 ants) with and without exposure to queen pheromones. Their analysis revealed many genes with differential expression or alternative splicing. Some of these genes, gene categories (GO and KEGG), or gene modules (groups of co-expressed genes) displayed similar direction of expression change across taxa, possibly indicative of “conserved effects on the transcriptome”. They also found that pheromone sensitive genes typically predate the divergence of ants and bees, have low connectedness, and often share particular characteristics (low gene body methylation, low expression levels, low codon bias, etc) in the honeybee. They also compared their gene modules with those from a previous study examining workers versus queens.

This study has generated a lot of data and the analysis appears sound. This study will be the basis for many useful follow up studies.

Thank you for your time and attention, and for the positive feedback.

While I really liked the idea of this study, I am not fully convinced that the main conclusion or interpretation (i.e., conserved effects on the transcriptome) is well supported by the results.

We agree that our original submission did not present enough data to fully support this claim, and so we added revised analyses that provide additional checks. In the revised manuscript, the conclusion that queen pheromones affect similar sets of genes in bees and ant is supported by:

- Figure 1C, which graphically illustrates that pheromones have similar effects on many orthologous genes. The formal statistics underlying this figure are shown in Figure 1D: there is a highly significant correlation across species.
- Figure 2, which shows that pheromone-sensitive genes are significantly enriched for many of the same GO and KEGG terms (using a better enrichment test than in the original manuscript).
- Figure 3, which shows that queen pheromone modulates three conserved modules of co-expressed genes in all 4 species; the associated statistics shows that we have no evidence for treatment-by-species effects.

1) The authors studied 2 ants (same genus) and 2 bees (the honeybee and the bumble bee) but the bumblebee showed almost no differences with pheromone treatment (1 gene for differential expression, Line 71; 0 genes for splicing, Line 81; Fig 1 left panel). One possibility is that there was not enough power (perhaps unknown factor associated with sample size, sub-optimal stage, pheromone dose/potency, etc) to detect the differences. In some ways, this is partially addressed in Fig 1 (right panel) by correlation in pheromone sensitivity across species. However, another possibility is that they don't really share the gene expression patterns, which appears to be the opposite of their main message.

Regarding the bumblebee, it is correct that there were comparatively few significantly differentially expressed genes, and that lower statistical power is probably to blame. However, variation in the number of statistically significant genes between species does not hamper our paper's main conclusions. This is because we purposely avoid drawing conclusions that

rest on whether any particular gene is significant or not, because the false negative rate is usually high in whole-transcriptome studies. Additionally, the false negative rate presumably varies between our 4 species due to differences in sample size and/or variance in the data (i.e. power), meaning that the smaller number of significant genes in bumblebees provides only weak evidence that the effects of pheromone differ across species. Indeed, Figures 1C, 1D, 2, and 3 illustrate that many/most of the effects are conserved in bumblebees, and we formally tested for species-by-treatment interactions, and found none.

2) Related, the overlap in gene expression or splicing is not different from random when comparing the honeybee (*Apis mellifera*) to either of the ants (Fig 1 Left panel; lines 97-99). To me, this does not seem to support conservation of a queen pheromone effect, and may indicate “neutral” patterns. Similarly, for the KEGG pathway analysis (Fig 2), it seems only the “ATP hydrolysis coupled proton transport” is up regulated in the controls for the honeybee plus the 2 ants (and no bumble bee, as indicated above). For GO (Fig 3), there are 2 categories (carbon metabolism and phototransduction in the fly).

It's important to note that our study (like pretty much all whole-transcriptome studies) has low power to detect differential expression for any particular gene, since our sample size is much lower than the number of genes (thousands of genes per test, $n = 44$ RNA-seq libraries), and we need to correct for multiple testing. To illustrate why this makes Venn diagrams a poor choice for measuring the degree of overlap, imagine that the true number of orthologous genes that are regulated by pheromones in both species A and species B is 100, but that the false positive rate when attempting to detect these 2×100 treatment effects was 40% (i.e. the statistical power was 60%, which is unusually high!). This means that our study would only detect 36 out of 100 overlapping genes on average (i.e. $100 \times 0.6 \times 0.6$). As explained in the Methods, this power issue is the reason that we presented several additional analyses to test for overlap (e.g. the correlations, the analysis of the top n -most pheromone sensitive genes, and the gene coexpression network analysis) in addition to the Venn diagrams. The same logic applies to the GO analyses, although we note that there are now several GO terms that are statistically significant in multiple species (see response to Reviewer 4 for details of the improved GO and KEGG analysis).

There was a paragraph describing this power issue (and how we tackled it using additional analyses) in the original Methods, but to highlight it further we have restructured the paper to mention power in the Results, and to clarify what conclusions can and cannot be drawn from the Venn diagrams.

Note, there was much more overlap between the 2 ants, but this is expected because they are in the same genus.

We agree this result is expected. An additional reason is that the two ant species have identical queen pheromones, which is not true for any of the other species pairs.

Other minor comments: 1) Figure 3 legend. “...for which at least two of the nine enrichment tests returned a significant result...” Should this be “one of the nine”? I may have missed something, but it seems some of the categories only have 1 significant results (e.g., RNA transport).

Many thanks for spotting this typo.

2) Lines 219 to 242. Here the authors discuss several genes which were somewhat highly ranked in their gene lists, but were not significant. And while I realize significance does not always need to meet a hard threshold, this study is based all on bioinformatics analysis. Perhaps complementary experiments would help?

We infer that the reviewer means experiments in which one knocks down a candidate pheromone-responsive gene and then tests for a phenotypic effect. We agree that these would be very interesting. Indeed, we are currently working on some of these in honeybees, in which RNAi experiments are difficult but possible (e.g. this preprint: <https://www.biorxiv.org/content/early/2018/05/04/3>). However, RNAi experiments are beyond the scope of the present study, whose aim is to determine how the whole transcriptome changes following experimental exposure to queen pheromone. Also, we note that robust functional genomics is extremely challenging in social

insects, and the first such studies are very recent (e.g. 10.1016/j.cell.2017.07.001); it would not be trivial to follow up our study with functional experiments.

Reviewer 2 comments

Summary

This study tests whether chemically similar compounds in social insect queen pheromones elicit similar effects on the transcriptome of different social insect taxa - namely, two species of ant and two species of bee. Specifically, they use RNA-Seq data from a previous study (Holman et al 2016) to identify transcripts differentially expressed upon exposure to species-appropriate queen pheromone.

Actually, our RNA-Seq data are newly-generated and are previously unpublished. We have made this clearer in the Methods.

From these data they test the idea that the response to pheromone ought to be ‘similar’ between species, and in the end draw this conclusion.

Our study addresses several other interesting and previously unexplored questions; we do not focus solely on testing for similarity. For example, we test whether queen pheromones mediate alternative splicing (for the first time), we identify and characterise the gene families targeted by queen pheromone (a first for ants and bumblebees, and an improvement over older microarray work in *Apis*), we test the hypothesis that the set of pheromone-sensitive genes overlaps with the set of genes that distinguishes queens and workers (a first for ants and bumblebees), and we show that queen pheromones primarily affect ‘ancient’ genes that originated before the origin of eusociality (another first). The revised Introduction hopefully makes this clearer: the last paragraph lists five hypotheses that we aimed to test in the present paper, only one of which concerns similarity.

Their main result is summarized in Figure 1, which they use to argue that i- all four species up-regulate ‘similar’ genes in response to queen pheromone (Fig 1 - top),

We hope the reviewers will agree from that the revised Figures 1-2 clarify the extent to which the patterns are similar across species. The conclusion that similarity is more common than difference is also supported by several of the statistical analyses.

ii- these genes are ‘similarly’ prone to splicing (Fig 1 - bottom),

Our original manuscript did not formally test for similarity in pheromone-sensitive splicing across species, but we think it is worthwhile doing, and therefore we added a new analysis which is described in the revised Results, and in Table S21 and Figure S6. In short, we found that genes whose isoform ratio is sensitive to queen pheromones tend to be the same genes in ants and bees, and are consequently enriched for somewhat similar KEGG and GO terms across species.

and iii- that these genes tend to show ‘similar’ pheromone sensitivity (Fig 1 - right).

Yes, that’s correct – see also the new Figure 1C, which shows this result more intuitively than in the original MS.

The authors bolden this narrative with other analyses, which include ‘similar’ KEGG (Fig 2) and GO (Fig 3) enrichment terms, evidence for ‘similar’ co-expressed gene modules across taxa (Fig 4) and, finally, a series of population genetics parameters that include codon-usage bias, positive selection and methylation tags (Fig 5) that they argue is evidence for pheromone-responsive genes being evolutionarily new (i.e., not ancient housekeeping genes) and having evolved in novel ways under (kin) selection.

We don’t think that we unjustifiably focus on any particular narrative. Our paper includes many of the standard follow-up analyses for a transcriptomics study (e.g. GO and KEGG

enrichment of the differentially expressed genes, and checking for overlap with earlier differential expression studies) in addition to some more novel ones (e.g. co-expression network analysis).

The data and analysis appear to be technically sound, and does include new and interesting results that may be of interest to the insect sociobiology community. As I highlight below, however, the main conclusion, which is conveyed in the title, is not obviously well supported nor, in some cases, are the results entirely novel.

Please see our reply to the editor regarding the appeal of our research, and see below for our responses to the comments regarding support for our conclusions and the novelty of our findings.

Major comments

Engagement with literature

L47 ‘To date only one study - in the honey bee *Apis mellifera* - has experimentally measured the effects of queen pheromone exposure on the transcriptome (Grozinger et al 2003)’. This statement, as written, does not appear to be correct. There are at least half a dozen other such studies, particularly from the Oldroyd group and Wenseleers group, and more studies yet from Grozinger/Robinson group. It was common in the microarray era (2003-2010) to examine transcriptional changes in honey bee worker gene expression upon exposure to queen pheromone. In addition, there are studies that did this on a smaller scale, using RT-PCR on a locus by locus basis.

To check whether we had indeed missed relevant literature, we conducted a formal literature review (see Appendix 1 at the end of this letter).

To summarise, we found a single additional study that applied queen pheromones and then measured gene expression across the whole transcriptome: “cGMP modulates responses to queen mandibular pheromone in worker honey bees” by Fussnecker, McKensie and Grozinger (2012, *J Comp Physiol A*). In our defence, this study is poorly cited, while the study we cited originally (Grozinger et al. 2003) has been cited c. 300 times. When reading the Fussnecker et al. paper, we were surprised to read that there was very little overlap in the lists of differentially expressed genes between these two studies (18 genes overlapped out of a possible >1000, which according to Fussnecker et al. is 3-fold less than expected by chance alone), even though the studies used similar methods and were conducted by some of the same researchers. This incongruity implies that one or both studies might not be robust, further increasing the value of the data in the present study. We have modified our Introduction to also cite Fussnecker et al, and to point out the difference in results. Lastly, we believe that RT-PCR studies of one or a few genes are not comparable to microarray or RNA-seq studies, since they do not measure the whole transcriptome and thus cannot answer many of our principle research questions (e.g. is there widespread pheromone-sensitive alternative splicing?).

Strong evidence for conclusions

L60 “Our principle aim was to test whether pheromone-sensitive genes, pathways, and transcriptional modules are similar or distinct across species”. This hypothesis of similar versus distinct is a difficult one to test because no doubt some aspects of the transcriptome response will be similar and other aspects will be distinct. Say for example, GO terms are 25% similar across taxa, or alternatively are 75% similar (and thus 25% distinct). In either case, what do you conclude? That they are similar... or that they are distinct? My point is that there is no *a priori* threshold or expectation to draw a conclusion either way and the author’s choice to emphasis ‘similar’ in this paper is arbitrary. A critic would ask: similar relative to what expectation? Maybe the four taxa chosen are this ‘similar’ for *any* chemical exposure (or treatment in general) based simply on phylogeny. If so, the chosen conclusion for conserved social evolution may have little to do with social evolution ...just phylogeny. I would therefore argue that in the category of ‘strong evidence for conclusions’ there is room for interpretation.

We agree that the term ‘similar’ was not used as carefully as it should have been. To address this comment, we made the new Figure 1C and Figure 2, and extensively re-drafted the manuscript with this feedback in mind. For example, we have changed the sentence quoted here to “Our principal aim was to measure the extent to which pheromone-sensitive genes, pathways, and transcriptional modules are similar or distinct across species”, which hopefully clarifies that we agree that similarity is both relative and subjective. Additionally, Figure 1C highlights that many genes are similar in their pheromone sensitivity across species, but there are also genes that respond only in one species (typically the honeybee – this species seems to have more pheromone-sensitive genes than the other three). The new Figure 2 shows that the patterns of GO and KEGG enrichment are largely the same across species. In sum, we think that there is a lot of similarity, but we cannot rule out the existence of some species-specific responses, and our new figures present the relative amounts of similarity and difference as clearly as possible.

Also, it is true that these species might display a similar response to other (non-pheromone) stimuli, given the shared ancestry of ants and bees (which diverged about 150MYA). However, we did not select a random stimulus: we are studying arguably the most important chemical for within-colony signalling. If the responses of ants and bees were conserved due to phylogeny (i.e. shared common ancestry followed by random genetic drift), this would still tell us something useful (namely that drift is astoundingly weak in social insects, since it has not produced much divergence despite 150MY of reproductive isolation between bees and ants, contradicting all of the evidence that social insects have small effective population sizes and thus have STRONG drift). However, we think it is very likely that the response of individuals to queen pheromone has been shaped by natural selection, since the receipt of queen pheromone alters many fitness-relevant traits (e.g. behaviour, fecundity, ageing, etc.). Thus, any similarities or differences between species seem very unlikely to result purely from non-adaptive ‘Brownian motion’ evolution.

Novel results

The single gene with the biggest expression was Major Royal Jelly Protein 3 (89-fold in QMP-). Indeed, five of the 12 top-most differentially expressed genes were MRJPs (MRJP 1, 2, 3 and 4, plus 3-like). The authors may wish to compare this result to a previous study that examined transcriptome-wide changes in worker gene expression upon exposure to pheromone (Thompson et al 2006. *Ins Mol Biol* 15:637-644). In that study, the authors likewise found multiple MRJP transcripts differentially expressed upon exposure to queen pheromone in *Apis mellifera*, including MRJP2 MRJP3 and MRJP4 as reported here (plus MRJP5 and MRJP7).

Thompson et al did not examine differential expression following exposure to queen pheromone; to quote them, “Our approach was to compare the gene expression profile of young wild-type workers against that of ‘anarchists’”. The paper does not mention pheromones except for one sentence in the Discussion. However, Thompson et al’s results are indeed relevant to our study, because ‘anarchist’ workers are more fertile (like workers deprived of queen pheromone). We therefore now cite this study when discussing our MRJP results.

Minor comments

L42 Authors should comment on what the ‘solitary common ancestor of the social Hymenoptera’ was or what that ancestral signal was. Was the signal part of a sexual signalling pathway prior to its co-option into reproductive division of labor?

By ‘solitary common ancestor of the social Hymenoptera’, we mean the most recent common ancestor of the extant social Hymenoptera, which (based on the principle of parsimony) is thought to have been a nest-building, wasp-like insect. However, we’re not sure it would be useful to add this to the Introduction, since it is tangential to our main study question and it is covered in the associated references about the phylogeny of Hymenoptera and the origins of eusociality.

Regarding the ancestral signal that evolved into the queen pheromone, we agree that a sex pheromone is indeed a likely possibility. We have added this hypothesis to the Introduction as suggested.

L51 ‘The chemical similarity of most species queen pheromones, coupled with the fact that queen pheromones influence similar phenotype traits in diverse lineages, leads us to predict...’ What are the 1- similar phenotypic traits and 2- diverse lineages that the authors are referring to?

We have re-written the Introduction to always spell out what is meant.

L62 What is meant by ‘targets’ of queen pheromones, and what are the mechanisms regarding ‘how’ signals operate that the authors are alluding to?

The revised manuscript spells out what was meant by “targets”: it now reads “We also wished to characterise the genes and pathways that respond to queen pheromone”. When we wrote “reveal **how** these fascinating signals produce their manifold phenotypic effects”, we were referring to the previous sentence, “Presumably, the profound changes in worker behaviour and physiology caused by queen pheromones are the result of pheromone-mediated changes in the transcriptome”. That is, our study sheds light on which genes are switched on and off following exposure to queen pheromone; these genes are presumably important in producing the observed phenotypic changes.

L65 What is meant by ‘shedding light’? Should spell-out a clear hypothesis to test.

The sentence now reads, “Fourth, we tested whether pheromone-deprived workers develop a more queen-like transcriptome to match their queen-like phenotype (e.g. laying eggs and living longer), thereby indirectly assisting the search for loci underlying caste dimorphism.” So, we predict that queen pheromone will repress the expression of queen-biased genes and/or stimulate the expression of worker-biased genes.

Figure 1- Note that in Fig 1 there really isn’t much overlap in the Venn diagrams, including no genes for *Bombus*. What proportion of Differentially Expressed Genes do these numbers represent? If there are more than 3000 shared orthologues between species (L130), then co-expressing between $n = 1$ to 5 of them among non-congeners (Fig 1 - top) or co-splicing between 1 to 2 of them (Fig 1 - bottom) does not seem to be very ‘similar’, as the authors conclude.

See our earlier arguments that Venn diagrams are a poor method to assess the similarity in changes in gene expression. Additionally, the lack of overlapping genes involving *Bombus* isn’t too surprising, because there was only a single statistically significant gene for *Bombus*.

The reviewer’s conclusion that 5 overlapping genes out of >3000 is not very much reflects an understandable misinterpretation of our original paper’s results, and so we have amended the Venn diagrams in Figure 1 for clarity (see end of this comment). Basically, to assess overlap of gene lists between species, one needs to ask how many of the genes that were significant in species A were also significant in species B. Not all of the >3000 orthologous genes were significant in at least one species, and not all of the genes that are significant in a single species have detectable orthologs in the other species – we did not make this very clear in the original figure.

To clarify what we mean via a worked example, for *Apis* honeybees and *L. niger* ants we had gene expression data for 4578 pairs of orthologous genes. Across these 4578 pairs of orthologs, 166 were significant in honeybees, 49 were significant in *L. niger*, and 3 were significant in both species. Given these results, the maximum number of overlapping genes that we could have observed given our power is 49 genes (not 4578), and the percentage of overlapping genes is $3/49 = 6\%$. Running a hypergeometric test, we see that this amount of overlap cannot reject the null hypothesis that the gene lists are independent ($p = 0.26$), though as we wrote above, this test is quite conservative and under-powered.

To address this comment, we have now added percentages (like the 6% in the previous example) to the Venn diagrams in Figure 1. The overlap is 6% for the two ants versus the honeybee, and 42% between the two ants, though we stress again that these numbers are very likely to be underestimates, since it is hard to get enough power to detect hundreds of true overlaps without a six-figure sequencing budget.

Figure 1 - Note too that the single asterisks denotes one case of overlap that was significantly higher than expected by chance (hypergeometric test, $p < 0.0001$), but this

overlap is between the two congeners, so the ‘chance’ is probably confounded by phylogeny.

We agree that it’s not unexpected that the effects of queen pheromones are more similar among related species relative to unrelated species. However, “confounded” may be the wrong word – our finding that the responses are more conserved within ants relative to the comparison of ants vs bees is a meaningful (though unsurprising) biological result, not a methodological or statistical artifact.

Figures 2 and 3 present important results, but this type of output is a bit prosaic: a list of KEGG and GO terms.

We have replaced these two figures with the new Figure 2, which contains more information, is based on an improved statistical test, and has better colours.

Figure 4 is perhaps the most interesting and novel result. The authors detect nine modules of co-expressed genes that contain between 38-1639 genes (Fig 4). Two of these modules significant predictors of pheromone treatment across taxa and are further shown to have low connectedness, which suggests they are expressed independently of the rest of the genome. This is an interesting and novel result.

We are glad that you agree. We think the coexpression network approach is a useful approach to study the structure of the transcriptome, and to increase statistical power by reducing its dimensionality – it does not make sense to test the 4000 genes one at a time, if those genes fall into a considerably smaller number of strongly coexpressed modules.

Figure 5- Gene-level measurements for *Apis mellifera*. Authors describe ‘up-regulation’ as a fold differences between queens and workers... but do the authors examine queens? I may have missed a subtle transition in the analysis, but in general the comparisons are between pheromone-treated and un-treated workers, so the introduction of queens was unexpected. In general, Fig 5 is hard to interpret.

The data comparing workers and queens in Figure 5 are from a prior study by Grozinger et al. (2007, Mol. Ecol.) – this was noted in the Methods, and we now also state it elsewhere in the paper. We used that dataset to test our prediction (given in the Introduction) that queen-biased and worker-biased genes should differ in their sensitivity to queen pheromone. To explain Fig 5 (now 4), the numbers give the correlations, the red and blue colours distinguish positive and negative correlations, and the asterisks indicate the p-values. The figure directs the eye to pairs of gene-level properties that are strongly correlated, as shown by bright red or bright blue for positive and negative correlations, respectively. We also spell out the results contained in the figure in prose in the Abstract, Results and Discussion, since the directions of some of the correlations might not be 100% clear from looking at the figure.

L 177 “Many genes were significantly affected by pheromone in two species, and (only) a single gene was significant in all three” : One could interpret this sparse pattern of minimal overlap to mean that the pheromone response is NOT particularly similar between species.

See earlier comments regarding Venn diagrams.

L178 - ‘Orthologous genes tended to show a similar level of pheromone sensitivity...’: Looking at Fig 1 where the correlations are provided (right panel), it doesn’t seem that similar to me. Spearman’s ρ is as low as 0.077 between *Lasius* and *Apis*.

The real, underlying correlation between genes (i.e. the correlation one would measure given infinite sample size) is likely to be higher than the values presented in Figure 1; possibly much higher. This is because our experiment estimates the pheromone sensitivity of individual genes with considerable measurement error, adding statistical noise that will tend to erode the overall correlation. Additionally, in the revision, we figured out how to calculate the similarity in GO term enrichment profiles between species, and found the correlations to be considerably higher (>0.4). This suggests that queen pheromones perturbed similar genetic pathways in each species. In response to this comment, we now write “The cross-species correlations are likely to be stronger than suggested by Figure 1D, because the sensitivity of

each gene is measured with error, adding noise to the underlying correlation.”

Reviewer 3 comments

The authors study patterns of differential gene expression in multiple species of social insects subject to treatments of queen pheromone. They find that different species show some similarities in the patterns of differential expression. In addition, genes differentially expressed tend to show certain characteristics.

Overall, I thought this was an interesting study that was well carried out. The manuscript is generally well written and the analyses are extensive and thought provoking. The study adds to our understanding of differential gene expression in social insect taxa.

Many thanks for your attention to our manuscript, and your positive appraisal.

Although this study was interesting and well conducted, I'm not sure that it would be broadly interesting to a wide audience.

Please see our reply to the editor regarding the appeal of the manuscript.

In addition, it is not so surprising that a pheromone putatively involved in the same function in different species would show at least some similar effects on gene expression in different taxa. So, although I viewed the study favorably, I was less sure about its general appeal.

As discussed in our Introduction, we don't think our result that queen pheromones have quite strongly similar effects across species was a foregone conclusion, for the following reasons:

- We don't know for sure that the pheromones we examined really have the same function in all 4 species. Although there is good evidence that all 4 of the pheromones we tested inhibit fecundity in workers, the *Apis* pheromone has many other documented effects (e.g. on worker behaviour) that have not yet been tested in the other 3 species. Also, bumblebee and ant queen pheromones are hypothesized to be involved in dominance signalling, but the same has not been said about honeybee queen pheromones.
- The pheromone treatments used different chemicals from distinct chemical families: the *Apis* pheromone is a blend of five keto acids and esters; the *Lasius* pheromone is a methylalkane, and the *Bombus* pheromone is a linear alkane.
- Bees and ants evolved their queen pheromones independently, and indeed they involved eusocial societies containing queens and workers independently, after their divergence around 150MYA. Even if the queen pheromones of bees and ants ultimately derive from a signalling system that was already present in their common ancestor, the re-purposing of this signal for queen-worker communication would represent convergent evolution, not shared ancestry.

Even if one disagrees that this result is interesting, we found a plethora of other novel and striking results. For example, the revised Figure 2 reveals the genetic targets of queen pheromones (for the first time, in ants and bumblebees, and for the first time using RNA-seq in honeybees), we show that these target genes existed before the origin of eusociality (a major unsolved puzzle), we demonstrated that queen pheromones mediate alternative splicing (a novel result in all 4 species), and we showed that queen pheromones target a non-random set of genes with regards to gene age, connectivity, DNA methylation, and expression dimorphism between queens and workers (again, a first).

Minor Comments

288 The authors apparently directly treated workers with queen pheromone. But it is not clear that workers ever come into direct contact with queen pheromone in natural settings in the way that it is applied in these experiments. That is, workers might detect queen pheromone if it is fed to them through trophallaxis. Or there might be queen pheromone spread in other non-tactile ways. But workers almost surely do not get queen pheromone

applied topically as seems to be the case in these experiments (although the methods were unclear on this). So the experimental effects of topical application of queen pheromone as obtained in this study may not be the same as those in natural conditions.

This comment stems from some ambiguous wording in our original submission. We did not topically apply the pheromone to workers: we agree that this would be unnatural. The pheromone treatment methods were fully described in our previous study that we referenced, and for brevity we did not duplicate them completely in the present manuscript. We have revised the manuscript to make it clearer that the pheromone was applied to the nest substrate, and not directly to the workers' bodies.

The authors look for and find many characteristics of genes differentially expressed under the pheromone treatments. I think it is certainly good that the authors did these analyses as the results are useful as presented. But I wouldn't take the results too far. For example, the findings that pheromone sensitive genes also tend to be unmethylated and differentially expressed in other contexts is as expected. Previous studies have shown that genes differentially expressed in one context tend to be differentially expressed in other contexts. And differential expression of genes in insects is also correlated with low methylation. So the point is that genes that are differentially expressed, regardless of the context, tend to show many other correlated properties. But it is unlikely the differential expression in the context of pheromones is causal of these results.

It's not clear what is meant by "I wouldn't take the results too far", because we were careful not to infer causality from a correlational result. For example, we do not claim that the difference in the frequency of DNA methylation between pheromone-sensitive and insensitive genes is proof that pheromones affect methylation (or that methylation changes a gene's response to pheromone). Regarding that correlation, we wrote "it is plausible that queen pheromones achieve their effects in part by altering the epigenome, and this hypothesis merits an experimental test", which we believe strikes an appropriately cautious tone. We also note a difference in opinion with Reviewer 4, who wrote "the authors are objective with their data".

We have re-read our manuscript to double-check for instances where we may appear to be over-reaching in our conclusions.

Reviewer 4 comments

Holman et al. analyse the transcriptomes of two ant and two bee species upon stimulation with a synthetic pheromone. The authors identify sets of differentially expressed genes in each species and analyse genes that tend to have similar responses upon pheromone stimulation in different species. They perform a comprehensive mining of conservation analyses, enriched sets of genes and co-expressed gene modules. They find that genes that are sensitive to pheromone stimulation tend to be present in the common ancestor of bees and ants. These genes are lowly expressed, have low connectedness and are hypomethylated. The authors argue that a module of genes, which was present in the common ancestor, is commonly stimulated in both ants and bees.

The introduction and the scientific questions are insightful. I appreciate that the supplementary material includes the R code to reproduce the supplementary figures, that there is a Github repository for all the code and that the authors are objective with their data. The downstream analyses of the differentially expressed genes are convincing and the conclusions of their analyses are interesting.

Many thanks for your attention to our manuscript. We are glad to hear that you thought our analyses were interesting and convincing, and that you agree that we were objective in drawing conclusions from our data.

However, the main conclusion is contradicted by some of their analyses. I provide more details below.

Please see our specific replies below.

Major points:

1. The RNA-seq data was generated from either whole bodies or lateral body halves. Is pheromone stimulation expected to have effects on the whole body of the animals or only on specific cell-types? I think it would be helpful for the authors to discuss if/how this could affect their analyses and conclusions.

Although data are scant, we agree that it is likely that pheromones (and indeed pretty much any stimulus) will affect some cell types differently to other cell types. Unfortunately there is essentially no information available about the tissue-specific transcriptome in social insects, so we cannot easily check our lists of genes against a tissue atlas (e.g. as a *Drosophila* researcher could do with FlyAtlas or modENCODE). However, the new Figure 2 gives a much better summary of the GO and KEGG results, providing hints as to which tissues showed the strongest effect (seemingly the ovaries, fat body, and nervous system, in line with predictions for a pheromone that affects fecundity and behaviour).

We also note that any differences in the response to pheromone between cell types would simply make it harder to detect responses that are restricted to certain cell types. For example, we might miss genes that only respond in the brain (just as studies of the brain transcriptome might miss genes that only respond in a specific region of the brain, for example). Given that this methodological issue is widely-recognised, and that we already remind the reader of the risk of false negatives, we did not add this point to the manuscript. We are open to further suggestions here.

2. The authors highlight in their title and discussion that transcriptomic changes upon pheromone stimulation are shared across species and tend to be associated with genes present in the common ancestor of bees and ants. I believe that this conclusion stems from the co-expression module analysis. However, this statement is contradicted by the data from Fig. 1: Despite some of the gene overlaps being higher than the overlaps expected by chance, the large majority of stimulated genes seem to be species-specific. For example, in Fig 1, *Lasius flavus* and *Lasius niger* share 29 genes, which is less than 10% of the genes stimulated in *L. flavus*. Moreover, the correlations shown in Fig 1 are positive but weak. Altogether, one could also reach the opposite conclusion that most pheromone-associated changes are species-specific.

See our earlier comments. To repeat in brief, the Venn diagram is very sensitive to false negatives in the data, such that genes that are borderline non-significant in species A, and either significant or borderline non-significant in species B, would not be detected as conserved pheromone targets. Also, the Venn diagram approach ignores information about the fold change in expression (since it reduces genes to being significant or not), which is why we prefer the correlations shown in the Figure 1 heatmaps. Additionally, our original Venn diagrams were misleading: the reviewer understandably inferred that less than 10% of genes overlapped in the two ant species, but the correct percentage (accounting for the fact that we did not detect orthologs for every gene, and the overlap can only be as large as the smallest number for the two component species) is 42% (the new Venn diagrams show the correct percentages). The correlations in the Fig 1 heatmaps are indeed a little low, but note that the fold change for each gene is measured with a lot of noise: if this noise were removed, the correlation would presumably go up. The conclusion that a large subset of genes show a conserved response is further strengthened by the co-expression module analysis, as the reviewer notes, as well as the GO and KEGG analyses.

3. To what extent are the differences between species in the Venn diagram in Fig. 1 due to thresholding effects (including the fact that *Bombus terrestris* has no significant genes)? It would be good to see more exploratory analyses in this respect; for example, it would be helpful to have a scatterplot of the log₂ fold changes (stimulated vs. control) between pairs of species, in which the points that were detected as significant in one species are highlighted in colour. This approach would provide information about genes with consistent fold changes across species.

We agree that ‘thresholding effects’ (i.e. results falling arbitrarily either side of $p < 0.05$) are important to consider, which is why we always aim to use statistical methods that depend on them as little as possible. We agree that ‘scatterplots of the log₂ fold changes (stimulated vs. control)’ are the correct way to measure similarity across species, and indeed this is basically

what we did to create which is now Figure 1D; although we quantified the strength of the correlations using Spearman's rank tests, rather than presenting the scatterplots. However, the scatterplots look ugly because there are around 4000 heavily over-plotted data points, and similar information is conveyed more succinctly in Figure 1D. To address this comment, we instead made the new Figure 1C, which gives an overview of the amount of similarity in pheromone sensitivity between each pair of orthologs in every species pair, as the scatterplots would.

4. To avoid that thresholding effects drive the conclusions, I would suggest that the authors re-analyse their data from the different species using linear models (for example multifactor models in edgeR or DESeq2, or ANOVA) or similar analyses. For example, a model $species + pheromone.condition$ would identify pheromone-stimulated across all samples, and a model with an interaction effect $species + pheromone.condition + species : pheromone.condition$ would identify species-specific effects.

We agree that it's important to be mindful of thresholding effects, and we made small improvements to a couple of our analyses with this comment in mind. For example, we now test for a difference in log-fold differential expression between conserved and lineage-specific genes using a Mann-Whitney test, instead of arbitrarily scoring genes as significant or not and then using a Chi-square test. Additionally, we don't think that trying a different software package would be a very useful check for robustness. This is because we selected the software EBSeq specifically because it seemed like a better choice for our study. The authors of the EBSeq software write that they specifically designed it to test for differential isoform expression in RNA-Seq experiments (as in our study), and it uses an empirical Bayes approach to deal with multiple testing, which is reputed to be an effective method. We suspect that competing software such as edgeR or DESeq2 would give very similar but distinct answers, but given that we selected EBSeq *a priori*, we would defer to EBSeq as the tiebreaker in any case where the results differed. Also, we do not wish to dilute readers' attention with an even longer list of results, which are repetitious of earlier results. Finally, we note that our analysis of the modules (Figure 3) already explicitly tests the species, pheromone treatment, and interaction effects. The associated results show that there is a significant treatment effect, and no significant Species:Treatment interaction, which might address the reviewers' concern.

5. In the gene-set enrichment analyses, it's not clear whether the p-values were corrected for multiple testing. Correcting for multiple testing is necessary, otherwise the authors can't draw conclusions from this analysis.

Motivated by this comment we did substantial research into the many different statistical tests that have been proposed for GO and KEGG enrichment, and learned a range of new analysis methods. If our reading of the literature is correct, it appears to be very common practice to present the p-values from GO analyses with no correction for multiple testing; we agree with the reviewer that this is problematic, since there are presumably many false discoveries. Part of the reason for the shortage of FDR testing is that it comes with some extra challenges in the context of GO analysis. Specifically, it is not trivial to calculate how strongly to correct the p-values because the different GO terms are non-independent. Another problem is that the number of GO tests that must be corrected for multiple testing, and thus the power of each test, depends on how well annotated the species is, and how many genes were successfully measured (in our study, this would introduce statistical artifacts by penalising results from well-annotated or well-sequenced subsets of the data). This latter issue does not affect the uncorrected p-values, eating into the benefits to inference that one gains from controlling the FDR.

Given all this complexity, we now present both corrected and uncorrected p-values in all figures and tables. If the reviewer knows of a better method than simply FDR-correcting all of the p values (naively with respect to the GO hierarchy), we'd be keen to hear it – as far as we could see, this is an unsolved problem in statistics.

Additionally, in the revised analysis, we switched to a different GO/KEGG enrichment method: Gene Set Enrichment Analysis. This method does not depend on classifying genes as either significant or non-significant; instead, one simply ranks the genes from most to least differentially expressed (e.g. using log fold change), and the test uses permutation to determine which GO terms are enriched among genes at the top and bottom of the list. This seems like an unambiguously better approach than the classic hypergeometric

GO/KEGG enrichment test, since it does not rely on an arbitrary p value threshold for differential expression, and it uses the full information content of the expression dataset (i.e. two significant genes with +2 or +100 fold-change in expression are not handled identically, as they are with the classic test). It also means that we are able to analyse all our species, including *Bombus*, which only had one significant gene and was thus not amenable to the hypergeometric GO/KEGG enrichment test.

Minor:

6. The accession number of the raw sequencing data does not exist in NCBI Biosamples.

Thanks for catching this: we have double-checked it, and the link is now working.

Appendix 1: Literature search for prior queen pheromone expression studies

Reviewer 2 wrote that, “There are at least half a dozen other such studies, particularly from the Oldroyd group and Wenseleers group, and more studies yet from Grozinger/Robinson group. It was common in the microarray era (2003-2010) to examine transcriptional changes in honey bee worker gene expression upon exposure to queen pheromone.”

This would mean that we were incorrect to claim that only one prior study experimentally manipulated queen pheromone, and then measured the whole transcriptome (Grozinger et al. 2003, *PNAS*). To verify the reviewer’s claim and find relevant papers we might have missed, we searched Web of Science on 19 July 2018, with the following search string:

((“apis mellifera”) AND queen AND pheromone AND (“gene expression” OR microarray OR RNaseq OR “RNA-seq”))

This search returned 95 hits, which we then read to determine whether they would invalidate our original claim that “To date only one study – in the honey bee *Apis mellifera* – has experimentally measured the effects of queen pheromone exposure on the transcriptome (Grozinger et al 2003)”. That is, we searched for experimental studies that manipulated the exposure to queen pheromone independently of other confounding factors, and then used a whole-transcriptome method of measuring gene expression such as microarray or RNA-seq. We found a single study that met this criterion which we had overlooked (Fussnecker et al, 2011), as mentioned in the reply to reviewer 2. The table on the following page gives the list of studies returned by our search, their suitability, and (in cases where this is not obvious from the title) the reasons why they do not invalidate our claim that there are only two prior studies examining the effect of queen pheromones on the transcriptome.

Table 1: Web of Science search hits.

Reference	Applied queen pheromone then measured the transcriptome?
Holt et al. (2018) Molecular, physiological and behavioral responses of honey bee (Apis mellifera) drones to infection with microsporidian parasites	No
Trawinski & Fahrbach (2018) Queen mandibular pheromone modulates hemolymph ecdysteroid titers in adult Apis mellifera workers	No
Mumoki et al. (2018) Reproductive parasitism by worker honey bees suppressed by queens through regulation of worker mandibular secretions	No
Song et al. (2018) Various Bee Pheromones Binding Affinity, Exclusive Chemosensillar Localization, and Key Amino Acid Sites Reveal the Distinctive Characteristics of Odorant-Binding Protein 11 in the Eastern Honey Bee, Apis cerana	No
Paris et al. (2018) Effects of the gut parasite Nosema ceranae on honey bee physiology and behavior	No
Nie et al. (2018) Comparative transcriptome analysis of Apis mellifera antennae of workers performing different tasks	No (nurses vs foragers)
Wu et al. (2017) Comparative transcriptome analysis on the synthesis pathway of honey bee (Apis mellifera) mandibular gland secretions	No (compares different castes)
Amsalem et al. (2017) Do Bumble Bee, Bombus impatiens, Queens Signal their Reproductive and Mating Status to their Workers?	No
Lopez et al. (2017) Cuticular hydrocarbon cues of immune-challenged workers elicit immune activation in honeybee queens	No
Teixeira et al. (2017) Proteomic analysis in the Dufour's gland of Africanized Apis mellifera workers (Hymenoptera: Apidae)	No
Villar & Grozinger (2017) Primer effects of the honeybee, Apis mellifera, queen pheromone 9-ODA on drones	No
Woyciechowski et al. (2017) Honeybee worker larvae perceive queen pheromones in their food	No
Traynor et al. (2017) Young and old honeybee (Apis mellifera) larvae differentially prime the developmental maturation of their caregivers	No
Kilaso et al. (2017) No evidence that DNA methylation is associated with the regulation of fertility in the adult honey bee Apis mellifera (Hymenoptera: Apidae) worker ovary	No

Table 1: Web of Science search hits.

Reference	Applied queen pheromone then measured the transcriptome?
Kilaso et al. (2017) DNA methylation of Kr-h1 is involved in regulating ovary activation in worker honeybees (Apis mellifera)	No
Nouvian et al. (2016) The defensive response of the honeybee Apis mellifera	No
Camiletti et al. (2016) A novel screen for genes associated with pheromone-induced sterility	No (it studies Drosophila)
Padilla et al. (2016) Chemical communication is not sufficient to explain reproductive inhibition in the bumblebee Bombus impatiens	No
Karpe et al. (2016) Identification of Complete Repertoire of Apis florea Odorant Receptors Reveals Complex Orthologous Relationships with Apis mellifera	No
Duncan et al. (2016) Notch signalling mediates reproductive constraint in the adult worker honeybee	No
Gruter & Keller (2016) Inter-caste communication in social insects	No
He et al. (2016) Starving honey bee (Apis mellifera) larvae signal pheromonally to worker bees	No
Mitaka et al. (2016) Caste-Specific and Sex-Specific Expression of Chemoreceptor Genes in a Termite	No
Holman et al. (2016) Queen pheromones modulate DNA methyltransferase activity in bee and ant workers	No (study by us)
Shpigler & Robinson (2015) Laboratory Assay of Brood Care for Quantitative Analyses of Individual Differences in Honey Bee (Apis mellifera) Affiliative Behavior	No
Galbraith et al. (2015) Reproductive physiology mediates honey bee (Apis mellifera) worker responses to social cues	No (qPCR of 1 gene)
Villar et al. (2015) Neurophysiological mechanisms underlying sex- and maturation-related variation in pheromone responses in honey bees (Apis mellifera)	No (qPCR of 1 gene)
Kuszevska & Woyciechowski (2015) Age at Which Larvae Are Orphaned Determines Their Development into Typical or Rebel Workers in the Honeybee (Apis mellifera L.)	No
Yan et al. (2015) DNA Methylation in Social Insects: How Epigenetics Can Control Behavior and Longevity	No
Weng et al. (2015) Binding interaction between a queen pheromone component HOB and pheromone binding protein ASP1 of Apis cerana	No

Table 1: Web of Science search hits.

Reference	Applied queen pheromone then measured the transcriptome?
McQuillan et al. (2014) Juvenile Hormone Enhances Aversive Learning Performance in 2-Day Old Worker Honey Bees while Reducing Their Attraction to Queen Mandibular Pheromone	No
Babis et al. (2014) Cuticular lipids correlate with age and insemination status in queen honeybees	No
Claudianos et al. (2014) Odor memories regulate olfactory receptor expression in the sensory periphery	No
Grozinger et al. (2014) From molecules to societies: mechanisms regulating swarming behavior in honey bees (Apis spp.)	No
Malka et al. (2014) Genomic analysis of the interactions between social environment and social communication systems in honey bees (Apis mellifera)	No (“We used microarrays to examine the MGs of virgin and mated queens, and queenright (QR) and queenless (QL) workers with or without activated ovaries”)
Mullen et al. (2014) Gene co-citation networks associated with worker sterility in honey bees	No (meta-analysis of earlier transcriptomic studies, all of which appear in this table)
Zhao et al. (2014) Sequence and expression characterization of an OBP1 gene in the Asian honeybee, Apis cerana cerana (Hymenoptera: Apidae)	No
Formesyn et al. (2014) Reproduction of honeybee workers is regulated by epidermal growth factor receptor signaling	No (RNAi experiment)
Manfredini et al. (2014) Molecular and social regulation of worker division of labour in fire ants	No (compares ants with and without a queen, not queen pheromone)
Uzunov et al. (2014) Swarming, defensive and hygienic behaviour in honey bee colonies of different genetic origin in a pan-European experiment	No
Camiletti et al. (2014) How flies respond to honey bee pheromone: the role of the foraging gene on reproductive response to queen mandibular pheromone	No
Koch et al. (2013) Caste-Specific Expression Patterns of Immune Response and Chemosensory Related Genes in the Leaf-Cutting Ant, Atta vollenweideri	No
Nino et al. (2013) Chemical Profiles of Two Pheromone Glands Are Differentially Regulated by Distinct Mating Factors in Honey Bee Queens (Apis mellifera L.)	No
Chan et al. (2013) Honey bee protein atlas at organ-level resolution	No
Geddes et al. (2013) Steroid hormone (20-hydroxyecdysone) modulates the acquisition of aversive olfactory memories in pollen forager honeybees	No
Richard & Hunt (2013) Intracolony chemical communication in social insects	No

Table 1: Web of Science search hits.

Reference	Applied queen pheromone then measured the transcriptome?
Chung-Davidson et al. (2013) An anti-steroidogenic inhibitory primer pheromone in male sea lamprey (Petromyzon marinus)	No
Holman et al. (2013) Are queen ants inhibited by their own pheromone? Regulation of productivity via negative feedback	No
Torto et al. (2013) Standard methods for chemical ecology research in Apis mellifera	No
McQuillan et al. (2012) Age- and behaviour-related changes in the expression of biogenic amine receptor genes in the antennae of honey bees (Apis mellifera)	No
Nino et al. (2012) Effects of honey bee (Apis mellifera L.) queen insemination volume on worker behavior and physiology	No
Eban-Rothschild et al. (2012) The Colony Environment, but Not Direct Contact with Conspecifics, Influences the Development of Circadian Rhythms in Honey Bees	No
Salomon et al. (2012) The role of tyramine and octopamine in the regulation of reproduction in queenless worker honeybees	No
Vergoz et al. (2012) Biogenic amine receptor gene expression in the ovarian tissue of the honey bee Apis mellifera	No (qPCR experiment comparing workers with and without a queen)
Fang et al. (2012) Differential antennal proteome comparison of adult honeybee drone, worker and queen (Apis mellifera L.)	No
Wang et al. (2012) Regulation of behaviorally associated gene networks in worker honey bee ovaries	No (compares two selected strains by microarray)
Kocher & Grozinger (2011) Cooperation, Conflict, and the Evolution of Queen Pheromones	No
Ament et al. (2011) Mechanisms of stable lipid loss in a social insect	No
Cardoen et al. (2011) Genome-wide analysis of alternative reproductive phenotypes in honeybee workers	No (microarray comparing self-selected fertile and sterile workers)
Jarosch et al. (2011) Alternative splicing of a single transcription factor drives selfish reproductive behavior in honeybee workers (Apis mellifera)	No
Richard et al. (2011) Effects of Instrumental Insemination and Insemination Quantity on Dufour's Gland Chemical Profiles and Vitellogenin Expression in Honey Bee Queens (Apis mellifera)	No
Fussnecker et al. (2011) cGMP modulates responses to queen mandibular pheromone in worker honey bees	Yes! This study contains a 2*2 factorial experiment, in which the authors present queen pheromone or a control, and cGMP or a control.

Table 1: Web of Science search hits.

Reference	Applied queen pheromone then measured the transcriptome?
Iovinella et al. (2011) Differential Expression of Odorant-Binding Proteins in the Mandibular Glands of the Honey Bee According to Caste and Age	No
Bloch & Grozinger (2011) Social molecular pathways and the evolution of bee societies	No
Feng et al. (2011) Antennal Proteome Comparison of Sexually Mature Drone and Forager Honeybees	No
Miklos & Maleszka (2011) Epigenomic communication systems in humans and honey bees: From molecules to behavior	No
Holman et al. (2010) Identification of an ant queen pheromone regulating worker sterility	No
Shemesh et al. (2010) Molecular Dynamics and Social Regulation of Context-Dependent Plasticity in the Circadian Clockwork of the Honey Bee	No
Shpigler et al. (2010) The transcription factor Kruppel homolog 1 is linked to hormone mediated social organization in bees	No (qPCR on one gene, in workers with and without a queen)
Kocher et al. (2010) The effects of mating and instrumental insemination on queen honey bee flight behaviour and gene expression	No
Wurm et al. (2010) Changes in reproductive roles are associated with changes in gene expression in fire ant queens	No
Kocher et al. (2010) Individual Variation in Pheromone Response Correlates with Reproductive Traits and Brain Gene Expression in Worker Honey Bees	No (compares workers that show a strong vs weak retinue response to queen pheromone)
Johnson (2010) Division of labor in honeybees: form, function, and proximate mechanisms	No
Vergoz et al. (2009) Peripheral modulation of worker bee responses to queen mandibular pheromone	No (qPCR on 3 genes)
Malka et al. (2009) The gene road to royalty - differential expression of hydroxylating genes in the mandibular glands of the honeybee	No (compares 2 genes in queens and workers)
Hasegawa et al. (2009) Differential gene expression in the mandibular glands of queen and worker honeybees, Apis mellifera L.: Implications for caste-selective aldehyde and fatty acid metabolism	No (qPCR of queens vs workers)
Beggs & Mercer (2009) Dopamine Receptor Activation By Honey Bee Queen Pheromone	No (physiological, not gene expression)
Fischer & Grozinger (2008) Pheromonal regulation of starvation resistance in honey bee workers (Apis mellifera)	No (qPCR of 4 genes)
Kocher et al. (2008) Genomic analysis of post-mating changes in the honey bee queen (Apis mellifera)	No (about queens)

Table 1: Web of Science search hits.

Reference	Applied queen pheromone then measured the transcriptome?
Le Conte et al. (2008) Quorum Sensing in Honeybees: Pheromone Regulation of Division of Labor	No
Richard et al. (2007) Effects of Insemination Quantity on Honey Bee Queen Physiology	No
Thompson et al. (2007) Experimental manipulation of ovary activation and gene expression in honey bee (Apis mellifera) queens and workers: Testing hypotheses of reproductive regulation	No (tests the response to CO ₂)
Wanner et al. (2007) A honey bee odorant receptor for the queen substance 9-oxo-2-decenoic acid	No (physiological, not gene expression)
Gotzek & Ross (2007) Genetic regulation of colony social organization in fire ants: An integrative overview	No
Grozinger et al. (2007) Uncoupling primer and releaser responses to pheromone in honey bees	No
Grozinger & Robinson (2007) Endocrine modulation of a pheromone-responsive gene in the honey bee brain	No (qPCR of one gene in workers with and without a queen)
Beggs et al. (2007) Queen pheromone modulates brain dopamine function in worker honey bees	No (qPCR of 3 genes)
Thompson et al. (2006) Towards a molecular definition of worker sterility: differential gene expression and reproductive plasticity in honey bees	No (see response to reviewer 2)
Katzav-Gozansky et al. (2006) Queen pheromones affecting the production of queen-like secretion in workers	No
Katzav-Gozansky et al. (2004) Queen-signal modulation of worker pheromonal composition in honeybees	No
Grozinger et al. (2003) Pheromone-mediated gene expression in the honey bee brain	Yes (cited in our manuscript)
Katzav-Gozansky et al. (2002) Honeybees Dufour's gland - idiosyncrasy of a new queen signal	No
Briand et al. (2001) Disulfide pairing and secondary structure of ASP(1), an olfactory-binding protein from honeybee (Apis mellifera L)	No
Katzav-Gozansky et al. (2000) Plasticity in caste-related exocrine secretion biosynthesis in the honey bee (Apis mellifera)	No
Danty et al. (1999) Cloning and expression of a queen pheromone-binding protein in the honeybee: an olfactory-specific, developmentally regulated protein	No

Reviewers' Comments:

Reviewer #2:

None

Reviewer #4:

Remarks to the Author:

In the revised version of the manuscript, all my previous comments were addressed, except for the one where I challenged the analysis used to conclude that "queen pheromones had similar effects on the transcriptome 234 in all four species." In this analysis, most of the overlaps of differentially expressed genes are either non-existent or very small, and the correlation of fold changes across species are very close to zero.

The authors attribute the low correlations to noise in their data and refused to provide plots of raw fold changes. These plots would enable the readers to judge by themselves how similar the transcriptome changes are to pheromone stimulation.

The authors responded that the low gene overlaps are due to the experiment's low power to detect differentially expressed genes. This is a good point, but one could easily get around the low-power issue by asking questions like "out of the genes that are differentially expressed in one species, for how many of them are fold changes consistent in the other three species?" Although some of the downstream analyses suggest that common effects might be present, it is unclear if this happens only for a few genes or for most stimulated genes.

In summary, it is not yet clear whether – or to what extent – the stimulated genes are species-specific or shared across species. A large part of the discussion is based on the assumption that most stimulated genes are shared across species, but I think the current analysis does not support this conclusion.

Response to the decision letter

Many thanks to the editor and reviewers for the valuable input and for your time.

The below response letter mostly addresses the concern that one of our conclusions is not supported by our data. We do this by:

- explaining and demonstrating the appropriateness of our statistical analyses,
- showing that our results are comparable to those of conceptually similar studies, and
- adding one new analysis which also supports our original conclusion.

Note also that the empirical claim we make about similarity is not as strong as the reviewers have inferred. For example, we opened the Discussion with the sentence “*To a **first approximation**, queen pheromones had **broadly similar** effects on the transcriptome in all four species*” (bold text added to indicate hedging). Thus, we concur with the reviewers that there is evidence for both similarity and difference in the data, though there is much more similarity than expected under the null hypothesis.

In the revised version of the manuscript, all my previous comments were addressed, except for the one where I challenged the analysis used to conclude that “queen pheromones had similar effects on the transcriptome in all four species.” In this analysis, most of the overlaps of differentially expressed genes are either non-existent or very small...

The following response assumes that “this analysis” refers to the Venn diagrams in Figure 1, since this is the analysis that counts the number of overlaps, and it is the only one of our analyses which does not obviously indicate conserved effects on gene expression.

In short, we think that the Venn diagram approach is very underpowered, as we explained at length in the MS and in the previous response letter, which is unfortunate because it is the most commonly-used method for investigating similarity in effects on gene expression in past studies. To recap, Venn diagrams are very likely fail to detect overlaps, and so the absence of many overlaps in the Venn is not evidence that the overlap is really absent (plus, we actually did observe several overlaps, in one case highly significant).

The limitation of the Venn approach stems from low statistical power. For example, if our power to detect a conserved, pheromone-sensitive gene in one species were on average 0.2, then the power to detect each gene that is expressed in all 4 species would be about $0.2^4 = 0.0016$. This means that we would fail to detect about 99.84% of the overlapping genes, producing a Venn diagram that looks the same as the one that we observed even if the real number of overlapping genes is very high.

Because that Venn-based result is under-powered, we produced several better-powered analyses and alternative plots of the data, which use a variety of means to test the hypothesis that the changes in gene expression are more similar than expected by random chance across species. All of them find evidence for similarity (often, strong similarity). These analyses include:

- The correlation in fold change across species (heatmap in Figure 1, plus the table below)
- The “eye” plot in Figure 1, which summarizes several independent analyses
- Overlap at the functional/pathway level in Figures 2-4
- Overlap at the level of co-expression modules of conserved genes in Figure 3-4
- The newly-added non-parametric test looking at the overlap in the similarity of the most pheromone-sensitive genes across species (see below).

The reviewer seems to disagree with some of these (see below), but they seemingly do not mention Figures 3 and 4. We think that it is highly unlikely that all of these tests are wrong and there is actually no similarity.

Finally, we note that the overlap indicated by the gene expression Venn diagram is not particularly low, despite the issue of low power. Our pairs of species had *at minimum* 6-42% of their differentially expressed genes in common. A lower-bound estimate of 42% similarity actually seems high, not “very small”, though similarity is a subjective term.

...and the correlation of fold changes across species are very close to zero.

We don't agree with the assertion that these correlations are “very close to zero” for several reasons. In the following response, we 1) show that they are not close to zero, and 2) illustrate that they are in fact similar or higher than the correlations seen in similar published studies to ours.

Here is the relevant table; the *rho* column describes the magnitude of the correlation (maximum possible value: 1), and the P column gives the results of a test with the null hypothesis that the true correlation is zero:

Species1	Species2	rho	p	sig
am	bt	0.136	3.232e-24	***
am	lf	0.100	1.478e-12	***
am	ln	0.077	1.853e-07	***
bt	lf	0.159	3.112e-39	***
bt	ln	0.127	2.282e-24	***
lf	ln	0.194	3.765e-61	***

The correlations are mostly in the range 0.1 – 0.2, and all of them are resoundingly non-zero according to the P-value. For something as noisy as gene expression, in a natural experiment on wild animals in which it was not possible to control all extraneous sources of variation, a correlation of 0.1 – 0.2 actually seems rather high.

Furthermore, this level of correlation between genes is typical of what other studies have found for other inter-species comparisons of strong effects on gene expression. We now give three examples of comparable recent studies published in competitive journals to illustrate this. Many more examples could likely be provided upon request.

Example 1

In another social insect example, Mikheyev and Linksvayer (*eLife*, 2014) found a $r = 0.14$ correlation between species for genes that show age polyethism in two ant species. Since this study is by one of us you might find this similarity unconvincing, and so we'll move on to studies by other people.

Example 2

In this week's issue of *PNAS*, Young et al. (2018) conclude that the changes in gene expression that characterise “monogamous” species are similar for several clades of vertebrates (birds, fish, frogs, mammals). The main evidence given for this conclusion is that there is a correlation in log-fold-

difference in gene expression between monogamous and non-monogamous species, between the clades of vertebrates. The raw log fold differences are given in Dataset S2 of Young et al, and we have plotted them here using the same presentation style as in our paper:

Figure for this review: Correlations between log fold differences values between clades. The log fold differences value indicates whether a gene is up or downregulated in the monogamous species in each clade, and by how much. Data from Young et al. 2018 *PNAS*. Note that these values are almost all lower than reported in the heatmap in our Figure 1.

As you can see, the correlations are closer to zero than ours, and most are not statistically significant.

Example 3

Our third example is taken from a within-species study that was carried out in a precisely-controlled laboratory system, and examined a single tissue type and a robust phenotypic difference (sex); here, one might expect the correlations to be substantially higher.

Vied et al. 2016 (*Journal of Comparative Neurology*) studied sex-specific gene expression in the hippocampus of lab mice (Vied et al. 2016, *Journal of Comparative Neurology*). In this study, the authors collected total RNA from the hippocampus of 3 males and 3 females for each of 6 different strains of lab mice. Their main research question was to test whether the genes with sex-specific expression are the same in all 6 mouse strains, or whether the set of sex-specific genes is unique to each strain. We would expect to see a strong positive correlation, because it is hypothesised that there is strong stabilising selection to maintain sex-specific expression and avoid unfit “intersex” phenotypes. It is also well-established that sex-specific hormones inhibit or stimulate particular suites of genes, and there is no reason to expect the targets of sex hormones to differ wildly between populations.

Vied et al did not provide log fold change values, so we downloaded their raw expression data, analysed differential expression using EB-Seq, and made a Venn diagram of their results (reassuringly, it is similar to the Venn in their paper). The Venn below shows that there are thousands of genes with significantly sex-specific expression, but almost all of these genes are unique to particular strains. Only 8 genes were sex-specific in all 6 strains, which initially seems very surprising.

Figure for this review: Venn diagram of statistically significant sex-specific genes in Vied et al, for 5 mouse strains. The Venn only shows 5 of the 6 strains, as one cannot easily plot a six-way Venn diagram.

Vied et al did not examine the correlations in log fold change, but we hypothesised that their Venn diagram might miss a lot of the similarity for the low-power reasons outlined above. We calculated the correlations, and again plotted them in a heat map. The correlation coefficients are similar in size to the ones we measured, despite the presumably lower amount of noise in Vied et al due to its being lab-based and intra-specific. This correlation matrix contradicts the main conclusion of Vied et al, that “only a few genes are differentially expressed across all of the strains”. **This illustrates that Venn diagrams can give biologically incorrect answers because of their low power.**

Figure for this review: Correlations in log fold sex difference in gene expression between each pair of the 6 mouse strains in Vied et al 2016.

So, why is it that the correlations in our study, and in all previous studies we could find, seem to be somewhat low? Is this real biology, a methodological artefact, or both?

As explained in the previous letter and in our MS (alluded to by the reviewer below), we think artefacts are very important. Essentially, we think that the fold change for each gene is measured with a considerable (and unknown) amount of error, which biases all of the correlations towards zero, meaning that the true correlations are very likely to be higher than the ones that we measured, and that the same is true for past studies like Vied et al.

To prove this and explain it graphically, we can perform do a simple exercise with correlated random numbers using R. Let's first generate a pair of vectors of correlated random numbers by sampling from a bivariate normal distribution. These numbers represent log fold change in gene expression for pairs of genes in two species, and in this example the true correlation coefficient is 0.5, both means are zero, and both variances are 1:

```
# Use a random number generator to make 5000 correlated random numbers
> x = mvtnorm::rmvnorm(n=5000, mean=c(0, 0), sigma = matrix(c(1, 0.5, 0.5, 1), ncol = 2))

# Plot each pair of numbers
> plot(x)
```

Compute the Pearson correlation coefficient

```
> cor(x[,1], x[,2])
[1]
[1,] 0.4930266
```

We can see that the scatterplot shows the expected moderately strong correlation, and the Pearson correlation coefficient is 0.5 as expected. Now, let's add some normally-distributed random measurement error to each observation, independently for both species:

Add some random noise to each number

```
x[,1] = x[,1] + rnorm(5000, mean = 0, sd = 2)
x[,2] = x[,2] + rnorm(5000, mean = 0, sd = 2)
```

```
> plot(x)
```

```
> cor(x[,1], x[,2])
[1]
[1,] 0.237593
```

The scatterplot now looks less convincing, and the Pearson correlation has dropped to 0.24; if we had added even more measurement error, it would have dropped further. Unfortunately, we cannot know the measurement error and so we cannot adjust for it, but because the measurement error is certainly greater than zero, we can be sure that the correlations in our table above are lower-bound estimates of the real correlations.

The authors attribute the low correlations to noise in their data and refused to provide plots of raw fold changes. These plots would enable the readers to judge by themselves how similar the transcriptome changes are to pheromone stimulation.

We did not “refuse to provide plots of raw fold changes”: we did add a new main-text figure showing the raw fold changes (Figure 1C – the “eye plot” below). This was explained in our response letter.

The reviewer asked for scatterplots, but as we explained, we think these are less informative since they only show 2 species at a time, and because each of the dots is misleading as explained in the previous example (unless we put error bars on each one of the thousands of points, but then it would be hard to read). We think this “eye plot” is a good way at showing the reading at a glance how many genes changed in all 4 species, and how many did not. As one can see there are many similarities and many differences (though again, some of the apparent differences could be false negatives due to low power to detect conserved changes in gene expression for any one gene).

However we are happy to provide the scatterplots as well (below). We think that they are an inferior way to present the data, since there is a lot of over-plotting and the axis limits are determined by a few outlying points. As a result, scatterplots provide little benefit over the rho statistics that we already presented (see the table given earlier, and the part D in the figure above). We now include these scatter plots as a new supplementary figure (Figure S3).

The authors responded that the low gene overlaps are due to the experiment's low power to detect differentially expressed genes. This is a good point, but one could easily get around the low-power issue by asking questions like "out of the genes that are differentially expressed in one species, for how many of them are fold changes consistent in the other three species?"

We agree – this is a good suggestion. We came up with a test that has a similar spirit, but does not rely on using the $p < 0.05$ threshold (which is arbitrary, and also raises practical problems because the number of significant genes varies between species).

Essentially the new test asks "If we rank the genes in species 1 by their pheromone sensitivity (measured by log fold change), how many of the top-n genes are also top in the top-n for species 2"? This test is a variant of a test that was already in the paper, which asked "Which genes are present in the top n-most pheromone-sensitive genes in multiple species". We again ran this new test for the top $n = 100, 200, \dots 500$ genes. The test works by randomly permuting the data to derive the null expected number of overlapping genes if the changes to gene expression occurred for independent, random genes in each species.

The new test's results are included as Table S9. In short, the table shows that there is statistically significant overlap for some but the following 3 out of the 6 possible species pairs: 1) *Apis* and *Bombus*, 2) the two *Lasius* species, and 3) *Bombus* and *Lasius niger*. The overlaps for 2 out of the remaining 6 species pairs (namely *Apis* with the two *Lasius* species) are almost significant ($p = 0.07$ and $p = 0.08$), leaving only one species pair (*Bombus* and *Lasius flavus*) where there p-value was not close to $p=0.05$. These results are not surprising, because this new permutation test is conceptually similar to the Spearman's rank correlation shown in the above table.

Although some of the downstream analyses suggest that common effects might be present, it is unclear if this happens only for a few genes or for most stimulated genes.

This comment appears to be a reference to Figure 2, shown here:

It is unclear why the reviewer thinks that the abundant similarity suggested by this plot (shown by the fact that most rows have the same colour) could be driven by just a handful of genes – that seems very implausible given the large number of diverse functions/pathways in the figure, and they don't explain why they reached that conclusion. Indeed, Tables S18 – S21 provide the numbers of enriched genes that underlie each of the enrichment results in Figure 2, and there are dozens/hundreds of genes in all, hopefully allaying the present concern.

In summary, it is not yet clear whether – or to what extent – the stimulated genes are species-specific or shared across species. A large part of the discussion is based on the assumption that most stimulated genes are shared across species, but I think the current analysis does not support this conclusion.

We hope that we have convinced you that our conclusions are well-supported. We believe that our approach is rigorous, up to the best standards in the field, and we stand by them.